# Bioactive Secondary Metabolites of the Genus *Diaporthe* and Anamorph *Phomopsis* from Terrestrial and Marine Habitats and Endophytes: 2010–2019

**DOI:** 10.3390/microorganisms9020217

**Published:** 2021-01-21

**Authors:** Tang-Chang Xu, Yi-Han Lu, Jun-Fei Wang, Zhi-Qiang Song, Ya-Ge Hou, Si-Si Liu, Chuan-Sheng Liu, Shao-Hua Wu

**Affiliations:** Yunnan Institute of Microbiology, School of Life Sciences, Yunnan University, Kunming 650091, China; xu2950129@163.com (T.-C.X.); luyihan1995@126.com (Y.-H.L.); wang_junfei@163.com (J.-F.W.); songzhiqiang1996@126.com (Z.-Q.S.); houyage@126.com (Y.-G.H.); liusisi1994@126.com (S.-S.L.); liucs313@126.com (C.-S.L.)

**Keywords:** ascomycetes, endophytic fungi, plant pathogens, biological activities, natural products

## Abstract

The genus *Diaporthe* and its anamorph *Phomopsis* are distributed worldwide in many ecosystems. They are regarded as potential sources for producing diverse bioactive metabolites. Most species are attributed to plant pathogens, non-pathogenic endophytes, or saprobes in terrestrial host plants. They colonize in the early parasitic tissue of plants, provide a variety of nutrients in the cycle of parasitism and saprophytism, and participate in the basic metabolic process of plants. In the past ten years, many studies have been focused on the discovery of new species and biological secondary metabolites from this genus. In this review, we summarize a total of 335 bioactive secondary metabolites isolated from 26 known species and various unidentified species of *Diaporthe* and *Phomopsis* during 2010–2019. Overall, there are 106 bioactive compounds derived from *Diaporthe* and 246 from *Phomopsis*, while 17 compounds are found in both of them. They are classified into polyketides, terpenoids, steroids, macrolides, ten-membered lactones, alkaloids, flavonoids, and fatty acids. Polyketides constitute the main chemical population, accounting for 64%. Meanwhile, their bioactivities mainly involve cytotoxic, antifungal, antibacterial, antiviral, antioxidant, anti-inflammatory, anti-algae, phytotoxic, and enzyme inhibitory activities. *Diaporthe* and *Phomopsis* exhibit their potent talents in the discovery of small molecules for drug candidates.

## 1. Introduction

*Diaporthe* is an important fungal genus of plant pathogens [1] belonging to the family Diaporthaceae, order Diaporthales, class Sordariomycetes [2]. It is mainly isolated from various hosts distributed in tropical and temperate zones and can cause diseases to a wide range of plant hosts, as well as humans and other mammals [3,4]. The ascomycetes of *Diaporthe* Nitschke 1870 and *Phomopsis* (Sacc.) Bubák 1905 are regarded to form a genus [5,6]. In Index Fungorum (2020), more than 1120 records of *Diaporthe* and 986 of *Phomopsis* are listed (http://www.indexfungorum.org/, accessed December 2020). There is a common understanding that, in these ascomycetes, the teleomorph states are named as *Diaporthe* and the anamorph states called as *Phomopsis* [7,8,9,10]. For a long time, a dispute has remained concerning whether the generic name should be defined as *Diaporthe* or *Phomopsis*. Due to the importance of this genus as plant pathogens, the classification of *Diaporthe* has been discussed by many researchers. Since *Diaporthe* was cited earlier and represents most of the species described in nature, more mycologists suggest that the use of *Diaporthe* as a generic name have more priority and is more suitable for the current study of this fungal group [11,12,13]. In recent years, the previous classification methods based on morphological characteristics are no longer applicable to the genus *Diaporthe* and advanced molecular techniques will replace them to solve the classification problem of *Diaporthe* [13,14]. In this review, we use the older name *Diaporthe* as the generic name.

Based on the existing literature investigations, more secondary metabolites have been separated from *Phomopsis* than *Diaporthe*. To date, a large number of compounds have been isolated from endophytic fungi of terrestrial plants in *Diaporthe* and *Phomopsis*, some of which originate from the marine environment (mainly mangroves and sediments). Most of compounds are classified as polyketides, which is the main structural type of secondary metabolites in this genus. The reported compounds showed various bioactivities, such as cytotoxic [15], antifungal [16], antibacterial [17], antiviral [18], antioxidant [19], anti-inflammatory [20], phytotoxic [21], and enzyme inhibition [22]. Up to now, there are 26 known species and various unidentified species of *Diaporthe* and *Phomopsis* have been studied for their metabolites. Our current review comprehensively summarize a total of 335 bioactive natural products from *Diaporthe* and *Phomopsis* between 2010 and 2019, covering their detailed chemical structures with classifications in structural types, as well as their bioactivities and habitats.

## 2. Bioactive Secondary Metabolites from *Phomopsis*

The *Phomopsis* fungi are important resource of bioactive compounds in the field of drug discovery, and have remarkable medical application value. According to the literature reports in recent ten years, a total of 246 bioactive compounds are summarized from *Phomopsis* herein. These substances have rich and diverse biological activities, such as cytotoxic, antifungal, antibacterial, antiviral, antioxidant, anti-inflammatory, phytotoxic, antimalarial, antialgae, antimigratory, pro-apoptotic, accelerating, and inhibiting the growth of subintestinal vessel plexus (SIV) branches, protecting effects on pancreatic *β*-cells, motility inhibitory and zoosporicidal potential, and enzyme inhibitory activities (Table 1). Among them, some interesting and promising bioactive compounds might be used in pharmaceutical and agricultural fields. The derived habitats of the *Phomopsis* strains can also be found in Table 1, which shows that there are 174 (accounting for 71%) and 66 (accounting for 27%) compounds obtained from terrestrial and marine environments, respectively, while six compounds (accounting for 2%) were not mentioned their habitats.

### 2.1. Polyketides

Polyketides are a large and diverse family of natural products, containing various chemical structures and biological activities [104]. In this review, 171 polyketides are summarized from *Phomopsis*, accounting for 70% of the total compounds from *Phomopsis*. The main bioactivities involve cytotoxic, antibacterial and antifungal activities. Herein, we classify these polyketides into xanthones, chromones, chromanones, benzofuranones, pyrones, quinones, phenols, oblongolides, and unclassified polyketides.

#### 2.1.1. Xanthones

Xanthones are a kind of compounds with the framework of 9H-xanthen-9-one, which mainly have anti-inflammatory, antimicrobial, antioxidant and cytotoxic activities [105]. A series of xanthones were obtained from the fermentation products of *Phomopsis* sp. isolated from *Paris polyphylla* var*. yunnanensis*, including three new compounds, 1,5-dihydroxy-3-hydroxyethyl-6-methoxycarbonylxanthone (**1**), 1-hydroxy-5-methoxy-3-hydroxyethyl-6-methoxycarbonylxanthone (**2**), 1-hydroxy-3-hydroxyethyl-8-ethoxy-carbonyl-xanthone (**3**), and seven known ones, pinselin (**4**), 1-hydroxy-8-(hydroxymethyl)-3-methoxy-6-methylxanthone (**5**), secosterigmatocystin (**17**), 1,7-dihydroxy-2-methoxy-3-(3-methylbut-2-enyl)xanthone (**22**), 1-hydroxy-4,7-dimethoxy-6-(3-oxobutyl)xanthone (**23**), asperxanthone (**24**) and 6-*O*-methyl-2-deprenylrheediaxanthone B (**25**). The cytotoxicities of all compounds to five human tumor cells (NB4, A549, SHSY5Y, PC3, and MCF7) were evaluated by using paclitaxel as positive control. The results showed that compounds **1** and **3** displayed cytotoxic activities and provided the IC_50_ values of 3.6 and 2.5 μM against A549 cells, and **1** gave an IC_50_ value of 2.7 μM against MCF7 cells. Compounds **22**–**23** showed weak activities and offered IC_50_ values greater than 10 μM for five tested cells. The others gave IC_50_ values between 3.8–10 μM against tested cells [23]. A new compound, 2,6-dihydroxy-3-methyl-9-oxoxanthene-8-carboxylic acid methyl ester (**6**), was isolated from *Phomopsis* sp. (No. SK7RN3G1) of mangrove sediment in the Shankou, Hainan, China. It showed cytotoxicity towards HEp-2 (IC_50_ = 8 μg/mL) and HepG2 (IC_50_ = 9 μg/mL) cancer cells [24]. Three secondary metabolites were characterized from fermentation products of *P*. *amygdali*, isolated from *Paris axialis*: 4,5-dihydroxy-3-(2-hydroxyethyl)-1-methoxy-8-methoxycarbonylxanthone (**7**), 1,8-dihydroxy-4-(2-hydroxyethyl)-3-methoxyxanthone (**8**), and paucinervin E (**13**). Compound **7** was active against A549 (IC_50_ = 2.6 μM) and PC3 (IC_50_ = 2.4 μM) cell lines. Compounds **8** and **13** displayed moderate activities with IC_50_ values in the range of 5.2–9.2 μM against one or more cell lines of NB4, A549, SHSY5Y, PC3 and MCF7 [25]. Hydroxyvertixanthone (**9**) was obtained from the endophytic fungus *Phomopsis* sp. YM 355364, originated from Chinese medicinal plant *Aconitum carmichaelii*. It showed antimicrobial activity with minimal inhibitory concentration (MIC) values of 256, 256, 128, and 64 μg/mL against *Escherichia coli*, *Bacillus subtilis*, *Pyricularia oryzae*, and *Candida albicans*, respectively [26]. The fermentation of fungus *Phomopsis* sp. derived from *Paris daliensis*, led to the isolation of six xanthones and identified as dalienxanthones A-C (**10**–**12**), 3,8-dihydroxy-4-(2,3-dihydroxy-1-hydroxymethylpropyl)-1-methoxyxanthone (**18**), oliganthins E (**19**), and cratoxylumxanthone D (**26**). These compounds were evaluated for cytotoxicities of five cancer cell lines (NB4, A549, SHSY5Y, PC3 and MCF-7). Compounds **12** and **18** were active to SHSY5Y with IC_50_ values of 3.8 and 3.5 μM, respectively, and the remaining compounds provided IC_50_ values in the range of 4.6–9.2 μM [27]. An investigation of extracts from fungus *P*. *amygdali* derived from the rhizome of *Paris polyphylla* var*. yunnanensis* afforded a new xanthone, 1,3-dihydroxy-4-(1,3,4-trihydroxybutan-2-yl)-8-methoxy-9*H*-xanthen-9-one (**14**). The bioactive results showed that **14** exhibited significant cytotoxic activity against A549 (IC_50_ = 5.8 μM) and PC3 (IC_50_ = 3.6 μM) [28].

An endophytic fungus *P. amygdali* associated with the rhizome of *Paris axialis* was cultured to obtain five xanthones: 3-methoxy-1,4,8-trihydroxy-5-(1ʹ,3ʹ,4ʹ-trihydroxybutan-2ʹ-yl)-xanthone (**15**), 8-methoxy-1,3,4-trihydroxy-5-(1ʹ,3ʹ,4ʹ-trihydroxybutan-2ʹ-yl)-xanthone (**16**), secosterigmatocystin (**17**), dihydrosterigmatocystin (**20**), and vieillardixanthone (**21**). The cytotoxic assay for NB4, A549, SHSY5Y, PC3 and MCF7 cancer cells were evaluated. The IC_50_ values of compound **15** against A549 and **16** against SHSY5Y were 3.6 and 4.2 μM, respectively. Compounds **17** and **20**–**21** displayed moderate activities with IC_50_ values in the range of 5.4–8.8 μM [29]. Studies of an endophytic fungus *Phomopsis* sp. (ZH76) from the stems of the mangrove tree *Excoecaria agallocha* contained a new *O*-glycoside compound, 3-*O*-(6-*O*-*α*-L-arabinopyranosyl)-*β*-D-glucopyranosyl-1,4-dimethoxyxanthone (**27**). The IC_50_ values of cytotoxicity for compound **27** on HEp-2 and HepG2 cells were 9 and 16 μmol/mL, respectively [30]. Phomoxanthone A (**28**), a dimeric tetrahydroxanthone, was extracted from *P*. *longicolla* of the mangrove tree *Sonneratia caseolaris*. Compound **28** had the strongest pro-apoptotic activity on human cancer cell lines and cisplatin-resistant cells, and its activity on healthy blood cells was reduced by more than 100 times. It was the most effective activator of mouse T lymphocytes, NK cells, and macrophages [31]. The study on secondary metabolites from fungus *Phomopsis* sp. IM 41-1 of mangrove plant *Rhizhopora mucronata* afforded phomoxanthone A (**28**) and 12-*O*-deacetyl-phomoxanthone A (**29**). When the concentration was 30 μg/ disk, compounds **28** and **29** showed moderate antimicrobial activities against *Botrytis cinerea*, *Sclerotinia sclerotiorum*, *Diaporthe medusaea*, and *Staphylococcus aureus*, but were inactive against *Pseudomonas aeruginosa* [32]. Four bioactive metabolites, dicerandrols A-C (**30**–**32**) and deacetylphomoxanthone B (**33**), were derived from *P*. *longicolla* S1B4. All compounds exhibited strong antibacterial activities against *Xanthomonas oryzae* KACC 10331. Dicerandrol A (**30**) also displayed notable antimicrobial activity against *S. aureus*, *B. subtilis,* and *C. albicans* with MIC values of 0.25, 0.125 and 2 μg/mL [34]. *Phomopsis* sp. HNY29-2B, isolated from mangrove plant *Acanthus ilicifolius*, produced four xanthone derivatives, **30**–**31**, **33** and penexanthone A (**34**). Compounds **30**–**31** and **33**–**34** displayed cyctotoxicities and provided IC_50_ values of 1.76–42.82 μM against MDA-MB-435, HCT-116, Calu-3, Huh7, and MCF-10A human cancer cell lines [35]. The structures of xanthones (**1**–**34**) are shown in Figure 1.

#### 2.1.2. Chromones

Chromones are a class of bioactive compounds with a benzo-γ-pyrone skeleton, which have been reported to have various activities, such as anti-tumor, anti-viral, antimicrobial, anti-inflammatory, and antioxidant [106]. *Phomopsis* sp. 33#, a mangrove endophytic fungus isolated from the bark of *Rhizophora stylosa*, produced four new chromone derivatives, (+)-phomopsichin A (**35**), (−)-phomopsichin B (**36**), phomopsichins C (**37**) and D (**38**), along with a known phomoxanthone A (**28**). These metabolites displayed low effects on inhibitions of acetylcholinesterase and *α*-glucosidase, radical scavenging function on DPPH and OH, and antimicrobial activities [33]. A cytotoxic chromone, chaetocyclinone B (**39**), was characterized from a culture of *Phomopsis* sp. HNY29-2B, an endophytic fungus obtained from the mangrove plant *A*. *ilicifolius* Linn. Compound **39** had cytotoxic activity against PC-3 (IC_50_ = 8.13 μmol/L) and DU145 (IC_50_ = 3.59 μmol/L) [36]. The fungus *Phomopsis* sp. IFB-ZS1-S4 isolated from *Scaevola hainanensis* Hance extracted a known pestalotiopsone F (**40**), which showed moderate inhibition on neuraminidase in vitro with IC_50_ value of 9.90 ± 0.42 μM [37]. Cultivation of *Phomopsis* sp. xy21 derived from the mangrove *Xylocarpus granatum* afforded a new xanthone-derived polyketide, phomoxanthone F (**41**). It showed inhibitory effects on VSV-G pseudotyped viral supernatant (HIV-1) with the inhibitory rate of 16.48 ± 6.67% at a concentration of 20 μM, which was higher than that of the positive control, efavirenz with a rate of 88.54 ± 0.45% [38]. 5-Hydroxy-3-hydroxymethyl-2-methyl-7-methoxychromone (**42**) was separated from the extracts of *Phomopsis* sp. (No. Gx-4) derived from mangrove sediment in ZhuHai, Guangdong, China. It showed low cytotoxic activity with IC_50_ values greater than 50 μmol/mL towards Hep-2 and HepG2. Moreover, it also significantly inhibited the growth of subintestinal vessel plexus (SIV) branches [39]. According to the bioassay-guided fractionation, two new chromones, phomochromones A (**43**) and B (**44**) were obtained from an endophytic fungus *Phomopsis* sp. of *Cistus monspeliensis*. They displayed remarkable antifungal, antibacterial, and antialgal activities against *Microbotryum violaceum*, *E. coli*, *Bacillus megaterium*, and *Chlorella fusca* [40]. Chemical investigation of *Phomopsis* sp. CGMCC No. 5416 isolated from *Achyranthes bidentata* led to the identification of two novel chromanones, phomochromanones A (**45**) and B (**46**). They showed anti-HIV activities with IC_50_ values of 20.4 and 32.5 μg/mL, and exhibited moderate cytotoxic activities towards A549, MDA-MB-231, and PANC-1 with CC_50_ values between 62.5–79.3 μg/mL [41]. A new naphtho-γ-pyrone compound, 5-hydroxy-6,8-dimethoxy-2-benzyl-4*H*-naphtho[2,3-b]-pyran-4-one (**47**), was obtained from *Phomopsis* sp. ZSU-H26 of the mangrove tree *E*. *agallocha*. This compound showed cytotoxic activity against HEp-2 (IC_50_ = 10 μg/mL) and HepG2 (IC_50_ = 8 μg/mL) [42]. The following work on the similar strain *Phomopsis* sp. (#ZSU-H76) from the same host additionally obtained phomopsis-H76 A (**48**), which significantly promoted the growth of the branches of SIV [43]. The structures of chromones (**35**–**48**) are shown in Figure 2.

#### 2.1.3. Chromanones

Chromanones have been widely studied due to their structural characteristics. They always have important biological and pharmacological activities, including cytotoxic, antimicrobial, antiviral, antioxidant, etc [107]. The culture of a marine fungus *Phomopsis* sp. (No. ZH-111) from mangrove sediment of Zhuhai, Guangdong, China, obtained a new isochroman, (3*R*,4*S*)-3,4-dihydro-4,5,8-trihydroxy-3-methylisocoumarin (**49**). It could promote the growth of SIV branches and exhibited low cytotoxic activity against Hep-2 and HepG2 cells with IC_50_ values above 50 mg/mL [44]. Three compounds were separated from *Phomopsis* sp. (No. Gx-4), including (3*R*,4*S*)-3,4-dihydro-8-hydroxy-4-methoxy-3-methylisocoumarin (**50**), 3,4-dihydro-8-hydroxy-3-methyl-1*H*-2-benzopyran-1-one-5-carboxylic acid (**51**), and 5,8-dihydroxy-4-methylcoumarin (**52**). All isolated compounds showed weak cytotoxic activities against Hep-2 and HepG2 cells with IC_50_ values above 50 μmol/mL. In addition, compounds **50** and **51** significantly promoted the growth of SIV branches, while **52** inhibited their growth [39]. The endophytic fungus *Phomopsis* sp. sh917 found in stems of *Isodon eriocalyx* var. *laxiflora* obtained (10*S*)-diaporthin (**53**), showing antiangiogenic activity that inhibited the angiogenesis process induced by vascular endothelial growth factor (VEGF) [45]. From agar-supported fermentation culture of *Phomopsis* sp. CMU-LMA derived from *Alpinia malacensis*, a trihydroxybenzene lactone, cytosporone D (**54**) was isolated. It showed antimicrobial activity and inhibited *E*. *coli* DnaG primase with an IC_50_ value of 0.25 mM [46]. Alternariol (**55**) and 5ʹ-hydroxyalternariol (**57**) were isolated from the endophytic fungus *Phomopsis* sp. A240 of *Taxus chinensis* var. *mairei*. Compound **55** showed low cytotoxicity against SF-268 (IC_50_ = 88.1 μM), MCF-7 (IC_50_ = 94.36 μM), and NCI-H460 (IC_50_ = 81.35 μM). Moreover, compound **57** had antioxidant activity with IC_50_ values of 42.83 μM [47]. Three compounds were sourced from *Endodesmia calophylloides* associated with *Phomopsis* sp. CAFT69, including alternariol (**55**), alternariol-5-*O*-methyl ether (**56**) and 5ʹ-hydroxyalternariol (**57**). In the range of 1–10 μg/mL, compounds **55**–**57** had certain motility inhibition and lytic activities on the zoospores of grapevine downy mildew pathogen *P*. *viticola* in dose- and time-dependent manner [48]. Phomochromanone C (**58**) was extracted from *Phomopsis* sp. CGMCC No. 5416. The bioactivity assay revealed that compound **58** showed cytotoxicity towards A549, MDA-MB-231, and PANC-1 with CC_50_ values of 69.4, 53.5, and 36.5 μg/mL, and it induced early apoptosis of PANC-1 cancer cells with the rate of 10.52% [41]. The structures of chromanones (**49**–**58**) are shown in Figure 2.

#### 2.1.4. Benzofuranones

Benzofuranones are an important intermediate of pharmacophores and drug molecules in natural products. Due to the furan ring being unstable and easy to open and crack, benzofuranones as a pharmaceutical intermediate have been widely concerned by pharmaceutical chemists [108]. The endophytic fungus *Phomopsis* sp. A123 isolated from mangrove plant *Kandelia candel* (L.) Druce, produced a novel depsidone, phomopsidone A (**66**), a known excelsione (**67**), and four known isobenzofuranones (**59**–**62**). All compounds showed different degrees of cytotoxicities against Raji and MDA-MB-435 tumor cells with IC_50_ values above 18 μM, displayed low antioxidant activities through DPPH radical scavenging effects, and exhibited antifungal activities [50]. The research on bioactive metabolites of marine fungus *Phomopsis* sp. (No. ZH-111) led to the isolation of 4-(hydroxymethyl)-7- methoxy-6-methyl-1(3*H*)-isobenzofuranone (**63**). Compound **63** inhibited the growth of SIV branches and exhibited low cytotoxic activity with IC_50_ values above 50 mg/mL against Hep-2 and HepG2 cells [44]. Chemical investigations of secondary metabolites from *Phomopsis* sp. BCC 45011 of *X. granatum* resulted in the identification of two known metabolites, cytosporones E (**64**) and P (**65**). Compounds **64** and **65** showed antimalarial activities against *Plasmodium falciparum* K1 with IC_50_ values of 2.02 and 3.65 μg/mL, and **64** exhibited cytotoxicity against MCF-7, NCI-H187, and Vero cells with IC_50_ values at 29.66, 5.84, and 4.53 μg/mL, respectively [51]. Cultivation of *Phomopsis* sp. CAFT69 afforded excelsional (**68**). In the range of 1–10 μg/mL, compound **68** had certain motility inhibition and lytic activities on the zoospores of grapevine downy mildew pathogen *P*. *viticola* in dose- and time-dependent manner [48]. Lithocarols A-F (**69**–**74**), with highly-oxygenated isobenzofuran skeleton, and isoprenylisobenzofuran A (**75**), were derived from *P*. *lithocarpus* FS508 isolated from a deep-sea sediment collected from the Indian Ocean. These metabolites were cytotoxic and provided IC_50_ values between 10.5–87.7 μM against HepG-2, MCF-7, SF-268, and A549 cells [52]. The endophytic fungus *Phomopsis* sp., separated from *Paris polyphylla* var. *yunnanensis*, gave three new arylbenzofurans (**76**–**78**) and four known compounds, moracin N (**79**), 2-(2′-methoxy-4′-hydroxy)-aryl-3-methy-6-hydroxybenzofuran (**80**), iteafuranal B (**81**), and moracin P (**82**). Compounds **76**–**82** showed inhibitory effects on tobacco mosaic virus (TMV) with inhibition rates of 18.6–35.2% [53]. The structures of benzofuranones (**59**–**82**) are shown in Figure 3.

#### 2.1.5. Pyrones

Pyrones are a kind of polyketides with six membered oxygen-containing heterocycles. As the precursor of many plants, animals, and microorganisms’ biosynthetic reactions, as well as its outstanding anti-tumor and antibacterial activities, researchers have shown strong interest [109]. Eight compounds were identified from the strain *P*. *asparagi* SWUKJ5.2020 isolated from medicinal plant *Kadsura angustifolia*, including five new 2-pyrone compounds, phomaspyrones A-E (**83** and **85**–**88**), along with three known metabolites, macommelin-8,9-diol (**84**), macommelin-9-ol (**89**), and macommelin (**90**). All isolated metabolites showed significant cytotoxic activities against six tested tumor cells (A549, Raji, HepG2, MCF-7, HL-60 and K562) with IC_50_ values of 1.0–26.8 μg/mL. However, phomaspyrone C (**86**) display better activity than the other compounds with IC_50_ values of 1.0–2.2 μg/mL against all tested cells [54]. The endophytic fungus *Phomopsis* sp. isolated from the plant *Cistus salvifolius*, yielded four new pyrenocines, pyrenocines J-M (**91**–**94**). They exhibited antibacterial and algicidal activities against *E. coli*, *B. megaterium*, and *C. fusca*. The antifungal assay showed that **92** and **94** were active against *M. violaceum*, and compounds **91**–**92**, and **94** were active against *Septoria tritici* [55]. An unusual pyrone metabolite, phomopsis-H76 C (**95**), was isolated from *Phomopsis* sp. (#zsu-H76), which inhibited the growth of SIV branch [43]. The structures of pyrones (**83**–**95**) are shown in Figure 3.

#### 2.1.6. Quinones

Quinones are natural bioactive molecules with unsaturated cyclic diketones, such as cytotoxic, antimicrobial, antiviral and anti-inflammatory activities. In recent years, the development of new anti-tumor quinones and their derivatives as lead compounds has become a hot topic [110,111]. Studies of the endophytic fungus *Phomopsis* sp. HCCB04730 associated with stems of *Radix Stephaniae Japonicae* obtained six known naphthoquinones **96**–**101**. These metabolites showed cytotoxic activities against A549, MDA-MB-231 and PANC-1 cancer cells with IC_50_ values of 1.1–120.5 μg/mL, and anti-HIV activities with IC_50_ values between 1.6–26.8 μg/mL [56]. Altersolanol B (**102**) was separated from *P*. *longicolla* HL-2232 of leaves of *Bruguiera sexangula* var*. rhynchopetala* collected from the South China Sea. Compound **102** showed antibacterial activity against *Vibrio parahaemolyticus* (MIC = 2.5 μg/mL) and *Vibrio anguillarum* (MIC = 5 μg/mL) [57]. A cytotoxic anthraquinone described as altersolanol A (**103**), was extracted from *Phomopsis* sp. (PM0409092) isolated from *Nyctanthes arbor*-*tristis*. Compound **103** had cytotoxic activity to 34 human cancer cells in vitro and gave the mean IC_50_ (IC_70_) value of 0.005 μg/mL (0.024 μg/mL) [58]. A new tetrahydroanthraquinone, named (2*R*,3*S*)-7-ethyl-1,2,3,4-tetrahydro-2,3,8-trihydroxy-6-methoxy-3-methyl-9,10-anthracenedione (**104**)**,** was separated from *Phomopsis* sp. PSU-MA214 associated with mangrove plant *Rhizophora apiculata*. Compound **104** was found to have low cytotoxic activity against MCF-7 and antibacterial activity against *S. aureus* ATCC25923 and methicillin-resistant *Staphylococcus aureus* SK1 [60]. The extraction of fungus *P. foeniculi* associated with *Foeniculum vulgare* in Bulgaria, resulted in the isolation of two octaketides anthracenones, altersolanols A (**103**) and J (**105**). They exhibited phytotoxic activities by leaf puncture bioassay [59]. Four known compounds were isolated from *Phomopsis* sp. derived from *Notobasis syriaca*, including 2-hydroxymethyl-4*β*,5*α*,6*β*-trihydroxycyclohex-2-en (**106**), (−)-phyllostine (**107**), (+)-epiepoxydon (**108**), and (+)-epoxydon monoacetate (**109**). All metabolites exhibited antifungal (*M. violaceum*), antibacterial (*E. coli, B. megaterium*), and algicidal activities (*C. fusca*), but **106** and **108** were inactive against *M. violaceum* [61]. A novel dihydronaphthalenone, phomonaphthalenone A (**110**), was derived from *Phomopsis* sp. HCCB04730. In terms of bioactive evaluation, compound **110** showed weak cytotoxic activity and moderate inhibitory activity on HIV with IC_50_ value of 11.6 μg/mL [56]. Ampelanol (**111**) was extracted from *Phomopsis* sp. HNY29-2B isolated from mangrove plant *A*. *ilicifolius*. Compound **111** showed antibacterial activity towards *B. subtilis* and *S. aureus* with MIC of 25 and 50 μM [62]. The structures of quinones (**96**–**111**) are shown in Figure 4.

#### 2.1.7. Phenols

Phenols are a kind of secondary metabolites which are widely distributed and have important physiological functions. They normally have antioxidant activity and play an important role in food industry [112]. Phomosine K (**112**) isolated from a *Phomopsis* strain showed remarkable antibacterial activity against *Legionella pneumophila* Corby and *E. coli* K12 [61]. Five known metabolites, phomosines A-D (**113**–**116**) and phomosine I (**117**) were isolated from a *Phomopsis* strain derived from *Ligustrum vulgare*. They had antibacterial and antifungal activities against *B. megaterium* and *M. violaceum*, except **116** was not active against *B. megaterium*. Moreover, compounds **113** and **116** inhibited the growth of algae [63]. Two new diphenyl ethers (**118**–**119**) were obtained from the culture of *P*. *asparagi* isolated from the rhizome of *Paris polyphylla* var. *yunnanensis*, collected in Kunming, Yunnan, China. These compounds displayed anti-methicillin-resistant *S. aureus* (anti-MRSA) activities with inhibition zone diameters (IZD) 10.8 ± 2.0 and 11.4 ± 1.8 mm, respectively [64]. Three new diphenyl ethers, 4-(3-methoxy-5-methylphenoxy)-2-(2-hydroxyethyl)-6-methylphenol (**120**), 4-(3-hydroxy-5-methylphenoxy)-2-(2-hydroxyethyl)-6-methylphenol (**121**), and 4-(3-methoxy-5-methylphenoxy)-2-(3-hydroxypropyl)-6-methylphenol (**122**), were extracted from *P*. *fukushii* of *Paris polyphylla* var. *yunnanensis*. Compounds **120**–**122** showed anti-MRSA activities and provided an IZD of 20.2 ± 2.5 mm, 17.9 ± 2.2 mm, and 15.2 ± 1.8 mm, respectively [65]. An endophytic fungus *P*. *fukushii*, separated from the rhizome of *Paris polyphylla* var. *yunnanensis*, gave three new isopentylated diphenyl ethers (**123**–**125**). Compounds (**123**–**125)** had notable anti-MRSA activities, and their IZD were 21.8 ± 2.4 mm, 16.8 ± 2.2 mm, and 15.6 ± 2.0 mm, respectively [66]. Two new diphenyl ethers (**126**–**127**) were obtained from the fermentation products of *P*. *fukushii* isolated from *Paris polyphylla* var. *yunnanensis*. The results of the anti-MRSA activities assay revealed that compounds **126** and **127** gave IZD of 13.8 ± 1.5 mm and 14.6 ± 1.6 mm, respectively [67]. Three new napthalene derivatives (**128**–**130)** were separated from *P*. *fukushii*, an endophytic fungus isolated from *Paris polyphylla* var*. yunnanensis*. Compounds **128**–**130** showed anti-MRSA activities with MCI values of 4, 4 and 6 mg/mL [68]. From fermentation products of the fungus *Phomopsis* sp. associated with *Paris polyphylla* var*. yunnanensis*, two new naphthalene derivatives (**131**–**132**) were obtained. Compounds **131**–**132** displayed anti-MRSA activities with IZD of 14.5 ± 1.2 and 15.2 ± 1.3 mm [69]. A culture of the marine fungus *P*. *lithocarpus* FS508 isolated from deep-sea sediment collected from Indian Ocean, obtained a new benzophenone, tenellone H (**133**). It showed cytotoxicity against HepG-2 (IC_50_ = 16 μM) and A549 (IC_50_ = 17.6 μM) [70].

The new metabolite, 16-acetoxycytosporone B (**134**), was sourced from *Phomopsis* sp. YM 355364 associated with *Aconitum carmichaeli*. In the bioassay, compound **134** had remarkable antifungal activity towards *C. albicans*, *Hormodendrum compactum*, and *Trichophyton gypseum* with MIC values of 32, 128, and 512 μg/mL [71]. Cultivation of *Phomopsis* sp. 0391 isolated from the stems of *Paris polyphylla* var. *yunnanensis* afforded cytosporone B (**135**) and dothiorelone A (**136**). These two compounds showed notable lipase inhibition and gave IC_50_ values of 115 and 275 μg/mL with Orlistat (IC_50_ = 43 μg/mL) as positive control [72]. Cytosporone B (**135**) was extracted from the cultivation of *Phomopsis* sp. PSU-H188, an endophytic fungus from *Hevea brasiliensis*. **135** showed protective effect on INS-1 832/13 pancreatic *β*-cells (EC_50_ = 11.08 μM) [73]. Two diastereomeric antineoplastic tenellone derivatives identified as lithocarpinols A (**137**) and B (**138**), were isolated from *P*. *lithocarpus* FS508, a deep-sea derived fungus derived from a sediment collected in the Indian Ocean. During the cytotoxic assay, compounds **137**–**138** showed inhibitory effects against HepG-2, MCF-7, SF-268, and A549 cancer cells with IC_50_ values ranging from 9.4 to 35.9 μmol/L [74]. Phomoindene A (**139**), a new indene derivative, was produced by *Phomopsis* sp. (No. GX7-4A) from the mangrove sediment of BeiHai, GuangXi, China. Compound **139** showed weak cytotoxicity againt KB, KBv 200, and MCF-7 cancer cells with IC_50_ values greater than 50 μmoL/mL [75]. Then, 4-Hydroxybenzaldehyde (**140**) was extracted from a strain of *Phomopsis* sp. YM 355364. The antimicrobial activities of **140** provided MIC values at 256 and 128 μg/mL against *B. subtilis* and *P. oryzae* [26]. An investigation of the extracts from *P*. *longicolla* HL-2232, afforded a new biphenyl derivative, 5,5′-dimethoxybiphenyl-2,2′-diol (**141**). Compound **141** displayed antibacterial activity against *V*. *parahaemolyticus* with MIC value of 10 μg/mL [57]. A known phenylethyl alcohol, phomonitroester (**142**), was derived from *Phomopsis* sp. PSU-MA214, exhibiting cytotoxicity with IC_50_ value of 43 μg/mL against KB [60]. Cytosporone U (**143**) was isolated from the fermentation products of *Phomopsis* sp. FJBR-11. This compound displayed inhibitory effect on TMV with IC_50_ value of 144.6 μg/mL [76]. Altenusin (**144**) was extracted from *Phomopsis* sp. CAFT69, possessing a certain motility inhibitory and lytic activity against the zoospores of grapevine downy mildew pathogen *P*. *viticola* between 1–10 μg/mL [48]. Cosmochlorins D (**145**) and E (**146**) produced by the endophytic fungus *Phomopsis* sp. N-125 of *Ficus ampelas*, showed significant cytotoxic activities against HL60 cells with IC_50_ values of 6.1 and 1.8 μM, and displayed growth-inhibition activities [77]. The structures of phenols (**112**–**146**) are shown in Figure 5.

#### 2.1.8. Oblongolides

Oblongolides are a kind of natural active products with novel norsesquiterpene γ-lactone. At present, oblongolides are relatively less reported than other kinds of polyketides. Most of them exist in the fungi of *Phomopsis*, and mainly have cytotoxic activities [113]. Three new oblongolides, oblongolides Z (**147**) and Y (**148**) and 2-deoxy-4*α*-hydroxyoblongolide X (**154**), were extracted from *Phomopsis* sp. BCC 9789 isolated from a wild banana (*Musa acuminata*) leaf. Compound **147** was found to have inhibitory effect on anti-herpes simplex virus type 1 (HSV-1) with IC_50_ value of 14 μM and showed cytotoxicities with IC_50_ values at 26–60 μM towards KB, BC, NCI-H187, and Vero cancer cells. Compound **148** was cytotoxic against BC (IC_50_ = 48 μM) and **154** showed anti-HSV-1 activity with IC_50_ value of 76 μM [78]. Five metabolites, oblongolides C1 (**149**), P1 (**150**), X1 (**151**), and C (**153**), along with 6-hydroxyphomodiol (**152**), were separated from the strain *Phomopsis* sp. XZ-01, an endophytic fungus of *Camptotheca acuminate*. Compounds **149**–**153** displayed different degrees of selective inhibition in cytotoxicities against HepG2 and A549 [79]. The structures of oblongolides (**147**–**154**) are shown in Figure 5.

#### 2.1.9. Unclassified Polyketides

Five compounds were obtained from *Phomopsis* sp. BCC 45011, including phomoxydiene C (**155**), 1893 A (**156**), mycoepoxydiene (**157**), deacetylmycoepoxydiene (**158**), and phomoxydiene A (**159**). All metabolites, except **156**, showed strong antimalarial activities against *P*. *falciparum* K1 with IC_50_ values at 2.41–3.52 μg/mL and cytotoxicities against KB, MCF-7, NCI-H187, and Vero with IC_50_ values between 1.49–45.5 μg/mL [51]. Seven new polyoxygenated cyclohexenoids, phomopoxides A-G (**160**–**166**) were obtained from the fermentation products of *Phomopsis* sp. YE3250 isolated from *Paeonia delavayi*. All compounds exhibited *α*-glycosidase inhibition with IC_50_ values from 1.47 to 3.16 mM, cytotoxic activities against Hela, MCF-7, and NCI-H460 cancer cell lines, and moderate antifungal activities against *C. albicans*, *Aspergillus niger*, *P. oryzae*, *Fusarium avenaceum*, and *H*. *compactum* [80]. A new geranylcyclohexenetriol, named phomentrioloxin (**167**), was obtained from *Phomopsis* sp. of the plant *Carthamus lanatus*. This compound showed phytotoxic activity and might be considered a potential mycoherbicide [81]. A new natural cyclopentenone, phomotenone (**168**) was produced by *Phomopsis* sp. Compound **168** displayed remarkable antifungal, antibacterial, and antialgal activities against *M. violaceum*, *E. coli*, *B. megaterium*, and *C. fusca* [40]. The cytotoxicity-guided investigation of the fungus *Phomopsis* sp. DC275 of *Vitis vinifera* yielded two new furanones, phomopsolidones A (**170**) and B (**171**), and a known phomopsolide B (**169**). All these metabolites showed weak phytotoxic and antibacterial activities [82]. The structures of unclassified polyketides (**155**–**171**) are shown in Figure 6.

### 2.2. Terpenoids

Terpenoids are a kind of natural bioactive substances with isoprene as scaffold, which are widely distributed and rich in species [114,115]. Herein, a total of 38 terpenoids, including three monoterpenoids, 25 sesquiterpenoids, seven diterpenoids, and three triterpenoids, were isolated from various *Phomopsis* strains, accounting for 15% of all the described metabolites, second only to polyketides. It is worth noting that some terpenoids showed interesting bioactivities, such as enzyme inhibitory and anti-inflammatory activities.

#### 2.2.1. Monoterpenoids

Monoterpenoids and their derivatives have a variety of biological activities, such as cytotoxic, antimicrobial, and anti-inflammatory, which have potential application value in clinical medicine [116]. Acropyrone (**172**) was extracted from culture of *Phomopsis* sp. HNY29-2B. Compound **172** showed antibacterial activity towards *B. subtilis* (MIC = 25 μM) and *P. aeruginosa* (MIC = 50 μM) [62]. A phytotoxic pentaketide monoterpenoid, nectriapyrone (**173**), was produced by the fungus *P*. *foeniculi* [59]. According to bioassay-guided procedure, a known compound, (1*S*,2*S*,4*S*)-trihydroxy-*p*-menthane (**174**) was obtained from *Phomopsis* sp., displaying antialgal activity against *C. fusca* and antibacterial activity against *E. coli* and *B. megaterium* [40]. The structures of monoterpenoids (**172**–**174**) are shown in Figure 7.

#### 2.2.2. Sesquiterpenoids

Sesquiterpenoids are the most abundant members of natural terpenoids because of their various structures and notable bioactivities. The chemical components of sesquiterpenoids had been found in plants, animals, microorganisms and marine organisms [117,118]. A series of sesquiterpenoids (**175**–**184** and **195**) were isolated from a strain of *Phomopsis* sp. TJ507A obtained from *Phyllanthus glaucus*. All compounds exhibited the inhibitory rates in the range of 19.4% to 43.8% against *β*-site amyloid precursor protein cleaving enzyme 1 (BACE1) at the concentration of 40 μM [83]. From the endophytic fungus *P. cassia* associated with *Cassia spectabilis*, two new diastereoisomeric cadinanes sesquiterpenes (**185**–**186**), (7*S*,10*R*)-3-hidroxicalamen-8-one (**187**), and aristelegone-A (**188**) were isolated. Compounds **185**–**188** showed antifungal activities towards *Cladosporium cladosporioides* and *Cladosporium sphaerospermum*, and acetylcholinesterase inhibitory activities [84]. Four metabolites were separated from *P*. *archeri* of *Vanilla albidia*, including three new sesquiterpenes, phomoarcherins A-C (**189**–**191**), and a known kampanol A (**192**). The cytotoxic activites of **189**–**192** provided IC_50_ values from 0.1 to 19.6 μg/mL against five cholangiocarcinoma cells (KKU-100, KKU-M139, KKU-M156, KKU-M213, and KKU-M214), and **189**–**190** showed little activities against the KB with IC_50_ values at 42.1 and 9.4 μg/mL. Compound **190** displayed antimalarial activity against *P. falciparum* (IC_50_ = 0.79 μg/mL) [85]. A new sesquiterpene, (+)-*S*-1-methyl-abscisic-6-acid (**193**), and a known (+)-*S*-abscisic acid (**194**), were extracted from *P*. *amygdali* of *Call midge*. Compounds **193**–**194** showed antibacterial activities against *P. aeruginosa* 2033E with MIC at 30 and 58 μg/mL [86]. Curcumol (**196**), isolated from *P*. *castaneae-mollissimae* GQH87 derived from medicinal plant *Artemisia annua*, showed cytotoxicity against MCF-7, HepG2, and A549 with IC_50_ values of 25.73, 65.18, and 178.32 μg/mL, respectively [87]. The cultivation of fungus *Phomopsis* sp. CAFT69, afforded two bioactive compounds, 9-hydroxyphomopsidin (**197**) and phomopsidin (**198**). Both of them showed motility inhibition and lytic activities on the zoospores of grapevine downy mildew pathogen *P*. *viticola* [48]. AA03390 (**199**) was isolated from a strain of *P*. *lithocarpus* FS508. The compound had low cytotoxicity with IC_50_ values of 25.5–29.6 μM against HepG-2, MCF-7, SF-268, and A549 [70]. The structures of sesquiterpenoids (**175**–**199**) are shown in Figure 7.

#### 2.2.3. Diterpenoids

Diterpenoids are a kind of terpenoids with various skeletons. They possess significant pharmacological activities, such as cytotoxic, antimicrobial, and anti-inflammatory activities [119]. A new diterpenes, libertellenone J (**200**), was derived from fungus *Phomopsis* sp. S12 isolated from *Illigera rhodantha*. This compound showed anti-inflammatory activity by reducing the production of NO, IL-1*β*, IL-6 and TNF-α, and inhibiting MAPKs and NF-κB pathways [88]. Four metabolites were extracted from *Phomopsis* sp. S12, including three new pimaranes, libertellenone T (**202**), pedinophyllols K (**203**) and L (**204**), together with a known compound, libertellenone C (**201**). Compounds **201**–**204** showed different degrees of anti-inflammatory activities against inhibiting the production of inflammatory factors (IL-1*β*, IL-6) by lipopolysaccharide in macrophages [89]. Secondary metabolites from fungus *P*. *amygdali* contained two known compounds, fusicoccin J (**205**) and 3*α*-hydroxyfusicoccin J (**206**). Biologically, compounds **205**–**206** showed antibacterial activities against *P. aeruginosa* 2033E with MICs at 26 μg/mL [86]. The structures of diterpenoids (**200**–**206**) are shown in Figure 8.

#### 2.2.4. Triterpenoids

Triterpenoids are a kind of organic compounds widely found in nature. They have attracted the attention of researchers because their structural diversity and rich bioactivities [120]. A new euphane triterpenoid, 3*S*,22*R*,26-trihydroxy-8,24*E*-euphadien-11-one (**207**), was isolated from *P*. *chimonanthi* obtained from medicinal plant *Tamarix chinensis* in the yellow river delta, Dongying. Compound **207** exhibited cytotoxicity against A549, MDA-MB-231, and PANC-1 cancer cells with IC_50_ values of 20.32, 19.87 and 30.45 μM, respectively [90]. The fungus *Phomopsis* sp. SNB-LAP1-7-32, occurring from plant *Diospyros carbonaria*, produced a first lupane-type triterpenoid, betulinic acid (**208**). Compound **208** displayed antiviral activity on inhibiting RNA-dependant RNA polymerase with IC_50_ values of 4.3 μM and cytotoxicity against HCT-116 and MRC-5 [91]. Oleanolic acid (**209**) was extracted from *P*. *castaneae-mollissimae* GQH87, which showed cytotoxicity against MCF-7, HepG2, and A549 with IC_50_ values of 16.61, 39.53, and 40.08 μg/mL, respectively [87]. The structures of triterpenoids (**207**–**209**) are shown in Figure 8.

### 2.3. Steroids

Steroids are secondary metabolites with a variety of chemical structures and biological activities. At present, many researchers try to find steroidal metabolites as potential lead compounds in drug design [121]. Till now, only nine steroids were isolated from *Phomopsis* and showed antifungal, anti-inflammatory, and antiviral activities. Five steroids were derived from culture of *Phomopsis* sp., an endophytic fungus separated from *A*. *carmichaeli*, including (14*β*,22*E*)-9,14-dihydroxyergosta-4,7,22-triene-3,6-dione (**210**), (5*α*,6*β*,15*β*,22*E*)-6-ethoxy-5,15-dihydroxyergosta-7,22-dien-3 one (**211**), calvasterols A (**212**) and B (**213**), and ganodermaside D (**214**). All isolated compounds displayed different degrees of selective antifungal activities against *C*. *albicans*, *A*. *niger*, *P*. *oryzae*, *F*. *avenaceum*, *H*. *compactum*, and *T*. *gypseum* with MIC values between 64–512 μg/mL [92]. Dankasterone A (**215**) and 3*β*,5*α*,9*α*-trihydroxy-(22*E*,24*R*)-ergosta- 7,22-dien-6-one (**216**) were isolated from *Phomopsis* sp. YM 355364. Compound **215** showed anti-influenza activity against H5N1pseudovirus (IC_50_ = 3.56 μM). Compounds **215**–**216** showed antifungal activities against *C*. *albicans*, *P*. *oryzae*, *H*. *compactum*, and *T*. *gypseum* with MIC values of 64–512 μg/mL [71]. A new functionalized ergostane-type steroid, named phomopsterone B (**217**), was obtained from *Phomopsis* sp. TJ507A isolated from medicinal plant *P*. *glaucus*. Compound **217** showed anti-inflammatory activity by inhibiting iNOS enzyme with an IC_50_ value of 1.49 μM [93]. Cyathisterol (**218**) was extracted from *Phomopsis* sp. YM 355364, displaying moderate antifungal activity toward *P. oryzae* (MIC = 128 μg/mL) [26]. The structures of steroids (**210**–**218**) are shown in Figure 9.

### 2.4. Macrolides

Macrolides are a class of medicinal compounds containing macrolactone ring structures, many of which are used as antifungal and antibacterial drugs in clinic, such as erythromycins [122]. Nowadays, a large number of macrolide antibiotics are widely used in the treatment of human diseases. Eight secondary metabolites were obtained from *Phomopsis* and showed cytotoxic, antimicrobial, and enzyme inhibitory activities. Three cytotoxic polyketides, Sch-642305 (**219**), LMA-P1 (**220**), and benquoine (**221**), were found in the endophytic fungus *Phomopsis* sp. CMU-LMA of *Alpinia malaccensis*. Compounds **219** and **221** also displayed antimicrobial activities [94]. The endophytic fungus *Phomopsis* sp. IFB-ZS1-S4 provided a known aspergillide C (**222**), which had moderate inhibitory effect on neuraminidase in vitro with IC_50_ value of 5.59 μM [37]. Four highly oxygenated tenellone-macrolide conjugated dimers, lithocarpins A-D (**223**–**226**), were obtained from *P*. *lithocarpus* FS508 isolated from the deep-sea sediment sample collected in the Indian Ocean. All metabolites (**223**–**226**) showed cytotoxic activities against three human tumor cells (SF-268, MCF-7, and HepG-2) with IC_50_ values in the range of 17.0–52.2 μM [95]. The structures of macrolides (**219**–**226**) are shown in Figure 10.

### 2.5. Alkaloids

Alkaloids are important nitrogen-containing organic compounds widely existing in microorganisms. At present, some alkaloids have been used to treat human diseases [123]. A total of 16 alkaloids have been isolated from *Phomopsis* and display various important bioactivities, such as cytotoxic, antibacterial, anti-inflammatory activities. Two compounds with special carbon skeleton, named phomopchalasins B (**227**) and C (**232**) were isolated from *Phomopsis* sp. shj2, an endophytic fungus obtained from the stems of *Isodon eriocalyx* var. *laxiflora*. Compound **232** showed cytotoxic activity against HL-60, SMMC-7721, and A-549 with IC_50_ values of 14.9, 22.7, and 21.1 μM, and displayed anti-inflammatory activity by reducing NO production (IC_50_= 11.2 μM). In addition, compounds **227** and **232** showed antimigratory activities against MDA-MB-231 with IC_50_ values of 19.1 and 12.7 μM [96]. Chemical investigation of *Phomopsis* spp. xy21 and xy22 obtained from leaves of the mangrove tree *X. granatum*, collected in Trang Province, Thailand, led to the isolation of a new cytochalasin, phomopsichalasin G (**228**). It showed cytotoxicities against HCT-8, HCT- 8/T, A549, MDA-MB-231, and A2780 cancer cells with IC_50_ values between 3.4–8.6 μM [97]. Three known compounds, namely 18-metoxycytochalasin J (**229**), cytochalasins H (**230**) and J (**231**), were obtained from *Phomopsis* sp. isolated from the nut of *Garcinia kola*. These three compounds exhibited cytotoxicities against HeLa (LC_50_ = 3.66–35.69 μg/mL) and Vero (LC_50_ = 73.88–129.10 μg/mL), and different degrees of antibacterial activities against six bacterial pathogens (*Vibrio cholera* SG24, *V*. *cholera* CO6, *V*. *cholera* NB2, *V*. *cholera* PC2, *Shigella flexneri* SDINT, and *S. aureus* ATCC 25923) [98]. The cytochalasins, epoxycytochalasin H (**234**) and cytochalasin N (**233**) and H (**230**), were extracted from *Phomopsis* sp. By254 derived from the root of *Gossypium hirsutum*. They showed remarkable antifungal activities with IC_50_ values between 0.1–50 μg/mL against *S. sclerotiorum*, *Bipolaris maydis*, *Fusarium oxysporum*, *B. cinerea, Bipolaris sorokiniana*, *Gaeumannomyces graminis* var. *tritici* and *Rhizoctonia cerealis* [99]. Cytochalasins H (**230**) and J (**231**), and alternariol (**55**) were extracted from *Phomopsis* sp. of *Senna spectabilis* and showed anti-inflammatory activities by inhibiting the production of reactive oxygen species (ROS). Compound **230** also showed antifungal and acetylcholinesterase enzyme (AChE) inhibitory activities [49]. Cytochalasin J (**231**) was derived from *P*. *asparagi* of plant *Peperomia sui* and exhibited antiandrogen activity (IC_50_ = 6.2 μM) [100]. The antibacterial diaporthalasin (**235**) was extracted from *Phomopsis* sp. PSU-H188, showing anti-MRSA activity with MIC of 4 μg/mL [73]. A phenylfuropyridone racemate, (+)-tersone E (**236**), and a known *ent*-citridone A (**237**), were separated from *P*. *tersa* FS441 derived from deep-sea sediment in the Indian Ocean. Compound **236** showed cytotoxicity with IC_50_ values at 32.0, 29.5, 39.5 and 33.2 μM towards SF-268, MCF-7, HepG-2, and A549 cancer cells. Compounds **236**–**237** had antibacterial activities against *S. aureus* with MIC value of 31.2 and 31.5 μg/mL [101]. Two new chromenopyridine derivatives, phochrodines C (**238**) and D (**239**) with 5*H*-chromeno[4,3-b]pyridine, were isolated from *Phomopsis* sp. 33# associated with the bark of *R*. *stylosa* in the South China Sea. Compounds **238**–**239** displayed anti-inflammatory activities with IC_50_ values of 49 and 51 μM by inhibiting nitric oxide production. Moreover, compound **239** also showed antioxidant activity with IC_50_ value at 34 μM [102]. A novel depsipeptide, PM181110 (**240**), was obtained from *P*. *glabrae* of *Pongamia pinnata*. It showed anticancer activity towards 40 human cancer cells in vitro (mean IC_50_ = 0.089 μM) and 24 human tumor xenografts ex vivo (mean IC_50_ = 0.245 μM) [103]. Fusaristatin A (**241**) was separated for the first time from *P*. *longicolla* S1B4, showing antibacterial activity against *X*. *oryzae* [34]. Exumolide A (**242**) from the strain *Phomopsis* sp. (No. ZH-111) significantly promoted the growth of SIV branches and showed low cytotoxic activity against Hep-2 and HepG2 [44]. The structures of alkaloids (**227**–**242**) are shown in Figure 11.

### 2.6. Flavonoids

Flavonoids are a kind of natural active substances of polyphenols. They are relatively less occurred in fungi [124]. In this review, only four flavonoids, quercetin (**243**) (Figure 12), luteolin (**244**), naringenin (**245**), and luteolin-7-*O*-glucoside (**246**) were isolated from *P*. *castaneae-mollissimae* GQH87. They displayed cytotoxic activities against MCF-7, HepG2, and A549 with IC_50_ values between 18.7 and 169.8 μg/mL [87].

## 3. Bioactive Secondary Metabolites from *Diaporthe* spp.

In the last ten years, a total of 106 bioactive secondary metabolites have been isolated from the genus *Diaporthe* (Table 2). These compounds exhibit various bioactivities, such as cytotoxic, antifungal, antibacterial, antiviral, antioxidant, anti-inflammatory, phytotoxic, antitubercular, antifibrotic, antidiabetic, antimigratory, antiangiogenic, antihyperlipidemic, inhibiting leishmanicidal, activating the NF-κB pathway, enzyme inhibition, inhibitory effects on osteoclastogenesis, antifeedant, contact toxicity, and oviposition deterrent activities. The habitats of the *Diaporthe* strains were also shown in Table 2, which revealed that there are 73 (accounting for 69%) and 32 (accounting for 30%) compounds isolated from terrestrial and marine environments, respectively, while only one compound (1%) was not mentioned with its habitat.

### 3.1. Polyketides

There are 67 polyketides reviewed from *Diaporthe* and they exhibit rich biological activities. Here, we classify these polyketides into the following structural types: xanthones, chromones, chromanones, furanones, pyrones, quinones, phenols, oblongolides, and unclassified polyketides.

#### 3.1.1. Xanthones

Chemical investigation of *Diaporthe* sp. SCSIO 41011 derived from mangrove plant *R*. *stylosa* led to identification of a known compound, 3,8-dihydroxy-6-methyl-9-oxo-9*H*-xanthene-1-carboxylate (**247**) (Figure 13). It showed influenza A virus (IAV) inhibition against A/Puerto Rico/8/34 H274Y (H1N1), A/FM-1/1/47 (H1N1), and A/Aichi/2/68 (H3N2) with IC_50_ values of 9.40, 4.80, and 5.12 μM, respectively [125]. Phomoxanthone A (**28**) with novel carbon skeleton was isolated from the fungus *Diaporthe* sp. GZU-1021 derived from a red-clawed crab *Chiromanteshaematochir* and *D*. *phaseolorum* FS431 of deep-sea sediment from the Indian Ocean. This compound showed anti-inflammatory activity by inhibiting nitric oxide (NO) production in RAW 264.7 cells with an IC_50_ value of 6.1 μM [126], and it displayed good cytotoxicity against MCF-7, HepG-2, and A549 with IC_50_ values of 2.60, 2.55, and 4.64 μM, respectively [127].

#### 3.1.2. Chromones

Chemical analysis of *Diaporthe* sp. GZU-1021 associated with *Chiromanteshaematochir* resulted in the identification of penialidin A (**248**) and (−)-phomopsichin B (**36**). They showed inhibitory effects on NO production with IC_50_ values at 11.9 and 16.5 μM [126]. Six bioactive metabolites were separated from *D*. *phaseolorum* SKS019 derived from mangrove plant *A*. *ilicifolius*, including four new compounds, (−)-phomopsichin A (**249**), (+)-phomopsichin B (**250**), diaporchromanones C (**251**) and D (**252**), along with two known compounds, (+)-phomopsichin A (**35**) and (−)-phomopsichin B (**36**). These metabolites showed moderate inhibition on osteoclastogenesis by inhibiting RANKL-induced NF-κB activation [128]. Pestalotiopsones F (**40**) and B (**253**) were isolated from *Diaporthe* sp. SCSIO 41011. The two compounds exhibited remarkable anti-IAV activities with IC_50_ values between 2.52–39.97 μM [125]. Two new benzopyranones, diaportheones A (**254**) and B (**255**), were extracted from *Diaporthe* sp. P133 derived from *Pandanus amaryllifolius*. They showed moderate antitubercular activities and provided MIC values of 100.9 and 3.5 μM against *Mycobacterium tuberculosis* H_37_Rv with Rifampin (MIC = 0.25 μM) as the positive control [130]. The structures of chromones (**248**–**255**) are shown in Figure 13.

#### 3.1.3. Chromanones

Two isocoumarins, (10*S*)-diaporthin (**53**) and orthosporin (**256**), were extracted from *D*. *terebinthifolii* LGMF907 isolated from *Schinus terebinthifolius*. They showed antibacterial activities against the methicillin-sensitive *Staphylococcus aureus* (MSSA) and methicillin-resistant *S. aureus* (MRSA) [131]. Cytosporone D (**54**) and mucorisocoumarin A (**257**) were isolated from the endophytic fungus *D*. *pseudomangiferaea* of *Tylophora ouata*. Compound **257** displayed anti-fibrosis activity with the inhibitory rate of 52.1% on the activation of human lung fibroblasts MRC-5 cells induced by TFG-*β* at 10 μM. Cytosporone D (**54**) showed cytotoxicity toward BGC-823 (IC_50_ = 8.1 μM), antioxidant activity with the inhibition rate of 63.3% by releasing MOA at the concentration of 10 μM, and moderate antidiabetic activity against protein tyrosine phosphatase 1B (PTP1B) [129]. The fungus *D*. *eres* derived from pathogen-infected leaf of *Hedera helix* produced an isocoumarin, 3,4-dihydro-8-hydroxy-3,5-dimethylisocoumarin (**258**), showing phytotoxic activity in *Lemna paucicostata* growth [132]. A novel metabolite, diportharine A (**259**), was obtained from the culture of *Diaporthe* sp. isolated from *Datura inoxia*. It showed notable antioxidant activity through DPPH radical scavenging effects (EC_50_ = 10.3 μM) [133]. The structures of chromanones (**256**–**259**) are shown in Figure 13.

#### 3.1.4. Furanones

Furanones are widely used in the field of synthesis, and the synthesized products have important pharmacological activities, such as antiviral, antitumor and antimicrobial [165]. Four bioactive furanones were derived from *Diaporthe* sp. SXZ-19 of *C*. *acuminate*, including the new (1*R*,2*E*,4*S*,5*R*)-1-[(2*R*)-5-oxotetrahydrofuran-2-yl]-4,5-dihydroxy-hex-2-en-1-yl(2*E*)-2-methylbut-2-enoate (**260**) and three linear furanopolyketides (**261**–**263**). These compounds had weak cytotoxicities against HCT 116 cells with the concentration at 10 μM [134]. A new 3-substituted-5-diazenylcyclopentendione, named kongiidiazadione (**264**), was separated from *D*. *kongii* of plant *C*. *lanatus*, which was phytotoxic component and showed low antibacterial activity against *Bacillus amyloliquefaciens* [135]. The structures of furanones (**260**–**264**) are shown in Figure 13.

#### 3.1.5. Pyrones

Four secondary metabolites were isolated from *D*. *maritima* of healthy *Picea mariana* and *Picea rubens* needles collected from the Acadian forest of Eastern Canada, including three dihydropyrones, phomopsolides A (**265**), B (**169**), and C (**266**), and a stable *α*-pyrone, (*S*,*E*)-6-(4-hydroxy-3-oxopent-1-en-1-yl)-2H-pyran-2-one (**267**). All compounds showed antifungal and antibiotic activities against *M*. *violaceum*, *Saccharomyces cerevisiae*, and *B*. *subtilis* [136]. Two known metabolites, 7-hydroxy-6-metoxycoumarin (**268**) and coumarin (**269**), were isolated from the endophytic fungus *D*. *lithocarpus* obtained from *Artocarpus heterophyllus*. Compounds **268** showed significant antifungal activity against *Sporobolomyces salminocolor* with the of 12.2 ± 0.3 mm, and **269** had a diameter inhibition zone of 12.3 ± 0.3 mm against the bacteria *B. subtilis* [137]. The structures of pyrones (**265**–**269**) are shown in Figure 13.

#### 3.1.6. Quinones

Two cyclohexeneoxidedione derivatives, phyllostine acetate (**270**) and phyllostine (**107**), showing strong antifeedant activities on *Plutella xylostella*, were extracted from culture of *D*. *miriciae* of plant *Cyperus iria*. Compounds **270** and **107** had the feeding inhibition of 100% at 50 μg/cm^2^ and the 50% feeding deterrence (DC_50_) values of 9 and 4.7 μg/cm^2^, displayed contact toxicities with the median lethal concentration (LC_50_) values of 4.38 and 6.54 μg/larva, and exhibited oviposition deterrent activities with the indexes of 100% and 28.6% at 50 μg/cm^2^, respectively [138]. The new biatriosporin N (**271**) was isolated from the marine-derived fungus *Diaporthe* sp. GZU-1021 and displayed anti-inflammatory activity by inhibiting NO production in RAW 264.7 cells with an IC_50_ value of 11.5 μM [126]. Two anthraquinone derivatives, emodin (**272**) and 1,2,8-trihydroxyanthraquinone (**273**), were isolated from an endophytic fungus *D*. *lithocarpus*. Emodin (**272**) exhibited notable cytotoxic activity against murine leukemia P-388 cells (IC_50_ = 0.41 μg/mL) and antibacterial activity against *B. subtilis*, *M. luteus*, *Pseudomonas fluorescences*, *E. coli*, and *S. cerevisiae* with the diameter of inhibition zones of 14.7, 13.2, 13.7, 12.7, and 11.7 mm, respectively. Compound **273** also displayed antibacterial activity against *B. subtilis*, *E. coli*, and *S. cerevisiae* at 14.2, 11.3, and 10.7 mm, respectively [137]. A bis-anthraquinone derivative, named (+)-2,2′-epicytoskyrin A (**274**), was isolated from *Diaporthe* sp. GNBP-10 of *Uncaria gambir* Roxb. It showed antifungal activity against 22 yeast strains and 3 filamentous fungi with MICs between 16–128 μg/mL [139]. Two cytoskyrin type bisanthraquinones, cytoskyrin C (**275**) and (+)-epicytoskyrin (**276**), were isolated from *Diaporthe* sp., an endophytic fungus obtained from *Anoectochilus roxburghii*. Compounds **275**–**276** could activate NF-κB pathway and increase the relative activity of luciferase at the concentration of 50 μM, and showed cytotoxicities against SMMC-7721 cells in dose-dependent manner [140]. The structures of quinones (**270**–**276**) are shown in Figure 14.

#### 3.1.7. Phenols

The phenolic metabolite, tyrosol (**277**), was extracted from *D*. *helianthi* isolated from *Luehea divaricate*. Tyrosol showed significant antagonistic activity against the tested pathogenic bacteria (*Enterococcus hirae*, *E. coli*, *M. luteus*, *Salmonella typhi*, *S. aureus*, and *Xanthomonas asc*. *Phaseoli*) [141]. 2,5-Dihydroxybenzyl alcohol (**278**) was derived from *D*. *vochysiae* LGMF1583 of medicinal plant *Vochysia divergens*, which showed cytotoxic activity against A549 (EC_50_ = 54.8 μM) and PC3 (EC_50_ = 9.45 μM) [143]. Four phytotoxic compounds, 4-hydroxybenzaldehyde (**140**), *p*-cresol (**279**), 4-hydroxybenzoic acid (**280**), and tyrosol (**277**), were isolated from *D*. *eres* of grapevine (*V*. *vinifera*) wood. In the leaf disk and leaf absorption bioassay, phytotoxicities of all compounds increased with the concentration ranging in 0.1–1 mg/mL [142]. Arbutin (**281**), obtained from an endophytic fungus *D*. *lithocarpus*, had moderate cytotoxicity against murine leukemia P-388 cells and gave an IC_50_ value at 2.91 μg/mL [137]. Two antibacterial metabolites, phomosines A (**113**) and C (**115**), were extracted from *Diaporthe* sp. F2934 of plant *Siparuna gesnerioides*. They were active against *S. aureus*, *M. luteus*, *Streptococcus oralis*, *Enterococcus fecalis*, *Enterococcus cloacae*, and *Bordetella bronchiseptica* with inhibition zone diameter from 6 ± 0.62 to 12 ± 1.18 mm at the concentration of 4 μg/μL [144]. Flavomannin-6,6ʹ-di-*O*-methyl ether (**282**) was extracted from an endophytic strain of *D*. *melonis* from *Annona squamosal*, which showed antimicrobial activity against *S. aureus* 25697, *S. aureus* 29213, and *Streptococcus pneumoniae* ATCC 49619 with MIC values of 32, 32, and 2 μg/mL, respectively [145]. Four secondary metabolites, acetoxydothiorelone B (**283**), and dothiorelones B (**284**), L (**285**) and G (**286**), were isolated from *D*. *pseudomangiferaea*. All of them displayed antifibrotic activities with the inhibitory rates of 17.4, 62.9, 59.2 and 41.1% on the activation of human lung fibroblasts MRC-5 cells induced by TFG-*β* at 10 μM, with pirfenidone (53.2%) as positive control at 1 mM [129]. Two diphenyl ether derivatives, diaporthols A (**287**) and B (**288**), were extracted from *Diaporthe* sp. ECN-137 isolated from the leaves of *Phellodendron amurense*. Compounds **287**–**288** exhibited anti-migration effects on TGF-β1-elicited MDA-MB-231 breast cancer cells with an concentration at 20 μM [146]. Tenellone C (**289**) was obtained from *Diaporthe* sp. SYSU-HQ3 of mangrove plant *E*. *agallocha*, displaying inhibitory effect on *M*. *tuberculosis* protein tyrosine phosphatase B (MptpB) (IC_50_ = 5.2 μM) [147]. Six compounds were isolated from endophytic fungus *Diaporthe* sp. SYSU-HQ3 derived from the branches of *E*. *agallocha*, including a new benzophenone derivative, tenellone D (**290**), four special 2,3-dihydro-1*H*-indene isomers, diaporindenes A-D (**291**–**294**), and isoprenylisobenzofuran A (**75**). All isolated compounds showed anti-inflammatory activities by LPS-Induced NO production in RAW 264.7 cells with IC_50_ values of 4.2–18.6 μM [148]. The structures of phenols (**277**–**294**) are shown in Figure 15.

#### 3.1.8. Oblongolides

Four lovastatin analogues, oblongolides D (**295**), H (**296**), P (**297**) and V (**298**), were obtained from the endophytic fungus *Diaporthe* sp. SXZ-19. These metabolites showed weak cytotoxic activities against HCT 116 cells with the concentration of 10 μM [134]. The structures of oblongolides (**295**–**298**) are shown in Figure 16.

#### 3.1.9. Unclassified Polyketides

Phomentrioloxin B (**299**) was obtained from a strain of *D*. *gulyae* isolated from *C*. *lanatus*, which had low phytotoxic effect to cause small necrosis against several weedy and crop plant species [149]. The fungus *Diaporthe* sp. SCSIO 41011 derived from mangrove plant *R*. *stylosa*, afforded two metabolites, *epi*-isochromophilone II (**300**) and isochromophilone D (**301**). Compound **300** displayed cytotoxicities against ACHN, OS-RC-2, and 786-O cells with IC_50_ values between 3.0 and 4.4 μM, and **301** had an IC_50_ of 8.9 μM against 786-O cancer cells [150]. The structures of unclassified polyketides (**299**–**301**) are shown in Figure 16.

### 3.2. Terpenoids

(1*R*,2*R*,4*R*)-Trihydroxy-*p*-menthane (**302**) was isolated from *Diaporthe* sp. SXZ-19, and displayed weak cytotoxicity on HCT 116 cells [134]. Two new *α*-pyrones, gulypyrones A (**303**) and B (**304**), were extracted from *D*. *gulyae*. Both of them showed phytotoxic activities and gulypyrone A caused necrosis against *Helianthus annuus* plantlets [149]. A pentaketide monoterpenoid, nectriapyrone (**173**), was isolated from culture of *D*. *Kongii*, showing phytotoxic activity [135]. A new brasilane-type sesquiterpenoid, diaporol R (**305**) was produced by an endophytic fungus *Diaporthe* sp. isolated from leaves of *R*. *stylosa*. Diaporol R had moderate cytotoxic effect on SW480 cancer cells and provided an IC_50_ value at 8.72 ± 1.32 μΜ [151]. Eremofortin F (**306**) was obtained from endophytic fungus *Diaporthe* sp. SNB-GSS10 of *Sabicea cinerea*. It showed cytotoxic activity against KB and MRC5 cells with IC_50_ values of 13.9 and 12.2 μM [152]. Two new eremophilanes, lithocarins B (**307**) and C (**308**), were extracted from *D*. *lithocarpus* A740, an endophytic fungus isolated from *Morinda officinalis*. These compounds displayed low cytotoxicities against SF-268, MCF-7, HepG-2, and A549 tumor cells with IC_50_ values between 37.68–97.71 μM [153]. The new triterpenoid, 19-nor-lanosta-5(10),6,8,24-tetraene- 1*α*,3*β*,12*β*,22*S*-tetraol (**309**), was obtained from *Diaporthe* sp. LG23 of the Chinese medicinal plant *Mahonia fortunei*, and displayed antibacterial activity against both Gram-positive and Gram-negative bacteria [154]. The structures of terpenoids (**302**–**309**) are shown in Figure 17.

### 3.3. Steriods

Only two steroids, 3*β*,5*α*,9*α*-trihydroxy-(22*E*,24*R*)-ergosta-7,22-dien-6-one (**216**) and chaxine C (**310**) (Figure 17), were isolated from *Diaporthe* sp. LG23, showing antibacterial activities against *B. subtilis* with streptomycin as a positive control [154].

### 3.4. Ten-Membered Lactones

Ten-membered lactones always have anti-tumor, anti-inflammatory, anti-viral, anti-bacterial and other pharmacological activities, exhibiting important medical value in clinical practice [166]. Phomolide C (**311**) from *Diaporthe* sp. of *Aucuba japonica* var*. borealis*, inhibited the proliferation of human colon adenocarcinoma cells with concentration of 50 μg/mL [155]. The endophytic fungus *D*. *terebinthifolii* GG3F6 derived from medicinal plant *Glycyrrhiza glabra*, afforded two known compounds, xylarolide (**312**) and phomolide G (**313**). Compound **312** had cytotoxicity in vitro against cancer cells MIAPaCa-2, HCT-116 and T47D cancer cells with IC_50_ values of 38, 100, and 7 μM and showed notable antimicrobial activity against *C. albicans* and *Yersinia enterocolitica* with IC_50_ values at 78.8 and 72.1 μM. Moreover, Compound **313** showed an IC_50_ value of 69.2 μM against *Y*. *enterocolitica* [156]. A novel metabolite, named xylarolide A (**314**), was isolated from the fungus *Diaporthe* sp. of *D*. *inoxia*. Compound **314** had remarkable cytotoxicities against MIAPaCa-2 and PC-3 cancer cells with IC_50_ values between 14–32 μM, and also showed antioxidant activity on DPPH radical scavenging effect (EC_50_ = 10.3 μM) [133]. The structures of four ten-membered lactones (**311**–**314**) are shown in Figure 17.

### 3.5. Alkaloids

18-Des-hydroxy cytochalasin H (**315**) was obtained from endophytic fungus *D*. *phaseolorum*-92C of *Combretum lanceolatum*. This compound inhibited leishmanicidal activity, displayed moderate antioxidant activity, and had cytotoxic activity against the breast cancer cells MDA-MB-231 and MCF-7 [157]. A series of the cytochalasins were extracted from *Diaporthe* sp. GDG-118 of *Sophora tonkinensis*, including 21-acetoxycytochalasins J_2_ (**316**) and J_3_ (**317**), 7-acetoxycytochalasin H (**319**), and cytochalasins J_3_ (**318**), H (**230**), J (**231**), and E (**320**). All isolated metabolites showed different degrees of antifungal activities against *Alternaria oleracea*, *Pestalotiopsis theae*, *Colletotrichum capsici*, and *Ceratocystis paradoxa* with MIC values of 1.56–100 μg/mL, and antibacterial activities against Gram-positive bacteria (*B*. *subtilis*, *B*. *megaterium* and *Bacillus anthraci*) and Gram-negative bacteria (*Proteus vuigaris*, *E*. *coli* and *Salmonella paratyphi B*) with MIC values in the range of 12.5–100 μg/mL [158]. The fungus *Diaporthe* sp. GZU-1021 yielded cytochalasin H (**230**) and 21-*O*-deacetyl-L-696,474 (**321**), which showed anti-inflammatory activities by inhibiting NO production in RAW 264.7 cells with IC_50_ values of 1.94 and 7.35 μM [126]. Cordysinin A (**322**) was derived from endophytic fungus *D*. *arecae* of *Kandelia obovate*. It showed anti-angiogenic activity against the human endothelial progenitor cells (EPCs) with IC_50_ value of 15.1 ± 0.2 μg/mL [159]. Further research led to the identification of 5-deoxybostrycoidin (**323**) and fusaristatin A (**241**) from *D*. *phaseolorum* SKS019 of mangrove plant *A*. *ilicifolius*. Compound **323** showed cytotoxic activity against MDA-MB-435 and NCI-H460 with IC_50_ values at 5.32 and 6.57 μM, and the IC_50_ value of **241** was 8.15 μM on MDA-MB-435 [160]. A new carboxamide, vochysiamide B (**324**), was extracted from new species *D*. *vochysiae* LGMF1583, which displayed antibacterial activity on the Gram-negative bacterium *Klebsiella pneumoniae* (KPC) with MIC value at 80 μg/mL and showed cytotoxic activity against A549 (EC_50_ = 86.4 μM) and PC3 (EC_50_ = 40.25 μM) [143]. Four compounds, diaporisoindoles A (**325**), B (**326**), D (**327**), and E (**328**), were obtained from an endophytic fungus *Diaporthe* sp. SYSU-HQ3. They all showed anti-inflammatory activities by reducing NO production with IC_50_ values of 22.7, 18.2, 8.9, and 8.3 μM, respectively [148]. Diaporisoindole D (**327**) also exhibited inhibitory activity towards *M. tuberculosis* protein tyrosine phosphatase B (MptpB) (IC_50_ = 4.2 μM) [147]. Phomopsin F (**329**) was isolated from *D*. *toxica*, and showed cytotoxic activity against HepG2 cells [161]. The structures of alkaloids (**315**–**329**) are shown in Figure 18.

### 3.6. Fatty Acids

Fatty acids are simple linear compounds that play an important role in the synthesis and catabolism of organisms [167]. Over here, six fatty acids are reported from *Diaporthe*. The fungus *D*. *phaseolorum* derived from *Laguncularia racemose*, afforded 3-hydroxypropionic acid (**330**), which showed antimicrobial activity against *S. aureus* and *S*. *typhi* [162]. A phytotoxic metabolite, 3-nitropropionic acid (**331**), was isolated from *D*. *gulyae*. Compound **331** was notably active in causing necroses on several weedy and crop plant species [149]. Two new fatty acids, diapolic acids A and B (**332** and **333**), were isolated from endophytic fungus *D*. *terebinthifolii*. They had moderate antibacterial activities against *Y*. *enterocolitica* with IC_50_ values of 78.4 and 73.4 μM [156]. Studies of the strain *Diaporthe* sp. JC-J7 from stems of *Dendrobium nobile* led to the isolation of a new compound, diaporthsin E (**334**). It showed low antihyperlipidemic activity on triglycerides (TG) in steatotic L-02 cells with the inhibition rate of 26% at the concentration of 5 μg/mL [163]. The novel anti-candidal metabolite, 3-hydroxy-5-methoxyhex-5-ene-2,4-dione (**335**), was derived from *Diaporthe* sp. ED2 of medicinal herb *Orthosiphon stamieus* Benth. It showed antifungal activity against *C*. *albicans* with MIC value of 3.1 μg/mL [164]. The structures of fatty acids (**330**–**335**) are shown in Figure 19.

## 4. Characteristics of Bioactive Secondary Metabolites from the Genus *Diaporthe* and Anamorph *Phomopsis*

In this paper, a total of 335 bioactive compounds from the genus *Diaporthe* and *Phomopsis* are summarized. There are 106 secondary metabolites from *Diaporthe* and 246 ones from *Phomopsis*, in which 17 compounds were obtained from both of *Diaporthe* and *Phomopsis*. These compounds are classified into polyketides, terpenoids, steroids, macrolides, ten-membered lactones, alkaloids, flavonoids, and fatty acids. As seen in Figure 20, about two thirds of all compounds reported from *Diaporthe* and *Phomopsis* are refered to polyketides, accounting for 63% and 70%, respectively. Moreover, terpenoids (8%, 15%), alkaloids (17%, 6%), and steroids (2%, 4%) were also produced by both of *Diaporthe* and *Phomopsis*. It is worth noting that fatty acids (6%) and ten-membered lactones (4%) are only reported from *Diaporthe*, while flavonoids (2%) and macrolides (3%) are only found in *Phomopsis*. Polyketides, as the largest member of the metabolites, are widely used in the field of medicine and play an important role in the treatment of cancer diseases.

The various bioactivities of the compounds isolated from *Diaporthe* and *Phomopsis* are presented in Figure 21, mainly containing cytotoxic, antibacterial, antifungal, antiviral, anti-inflammatory, antioxidant, antialgae, enzyme inhibition, and phytotoxic activities. Most of compounds have at least one kind of bioactivities. As seen in Figure 21 and Table 1 and Table 2, secondary metabolites of *Diaporthe* and *Phomopsis* mainly exhibit cytotoxic, antibacterial and antifungal activities, accounting for 73% of all compounds, with 56 in *Diaporthe* and 200 from *Phomopsis*. Interestingly, in recent years, more and more compounds with anti-inflammatory, antioxidant and enzyme inhibitory activities have been studied in important human diseases.

## 5. Conclusions

This review presents the diverse chemical structures and bioactivities of 335 compounds isolated from 26 known species and various unidentified species of the genus *Diaporthe* and its anamorph *Phomopsis* between 2010–2019. Here, we can see from Table 1 and Table 2, among all of the reported compounds, there are 236 (accounting for about 70%) and 92 (about 27%) compounds derived only from terrestrial and marine environments (including mangroves, sediments, deep-sea fungi and marine animals), respectively. In addition, only one compound is obtained from both of terrestrial and marine environments. In contrast, six compounds are not mentioned with their habitats in the literature. Polyketides represent the main chemical population, accounting for 64%. About 73% of all metabolites possess cytotoxic, antibacterial, and antifungal activities. The species named as *Phomopsis* significantly produce much more compounds than *Diaporthe*, and most strains have not yet been identified at the species level. In conclusion, these results illustrate that the metabolic resources of *Diaporthe* and *Phomopsis* are of great value and deserved to conduct further research. Interestingly, in the past three years, there have been more reports on the secondary metabolites of the fungi in *Diaporthe* and *Phomopsis* than before, displaying an increasing trend, which indicates that *Diaporthe* and *Phomopsis* are regarded as important sources for discovering new natural bioactive substances.

In the past many years, lots of interesting fungal bioactive metabolites had been widely developed into new drugs, like antibiotics. Although most compounds obtained from *Diaporthe* and *Phomopsis* fungi had been studied on their isolation, structures, and activities, the in-depth research on pharmacological mechanisms and development of potent active compounds in drugs are still less. According to current studies, some compounds with remarkable bioactivities may serve as potential drug candidates in the future, such as cytotoxic altersolanol A and PM181110, and antimicrobial dicerandrol A. In order to ascertain the therapeutic potential of these compounds, further studies of pharmacological and producing mechanisms are required.

The fungal species in *Diaporthe* and *Phomopsis* have been considered to be important sources that can produce diverse and novel bioactive metabolites, which has attracted many natural product chemists and pharmacologists to study in recent years. The metabolites produced by *Diaporthe* and *Phomopsis* have rich biological activities, which is enough to show the importance of its metabolic resources. Nowadays, many fungi produce interesting bioactive metabolites that have been studied for their biosynthesis pathway, while similar studies in *Diaporthe* and *Phomopsis* are performed relatively less often. In the following work, the microbial biosynthesis pathway might be considered for further developing valuable products from *Diaporthe* or *Phomopsis*, which are hoped to be used as drug molecules for disease treatment. However, it cannot be ignored that *Diaporthe* or *Phomopsis* are important plant pathogens which might cause a wide range of plant host diseases and even serious human pathogens. In the future work, we should also focus on the role of metabolites produced by these pathogens, as well as the relationships with their hosts.

## Figures and Tables

**Figure 1 microorganisms-09-00217-f001:**
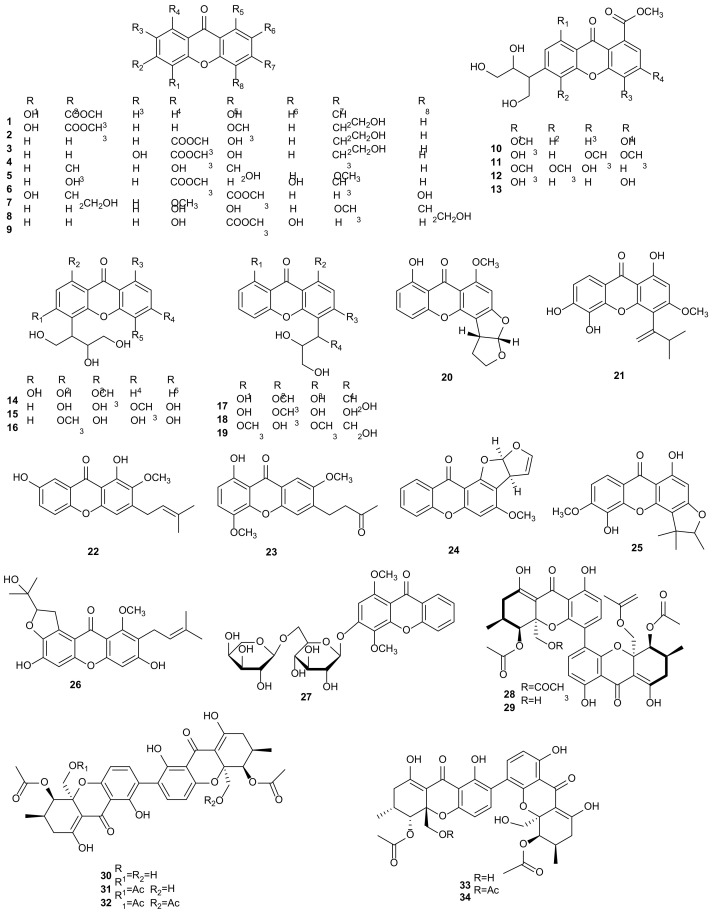
Chemical structures of compounds **1**–**34** from *Phomopsis*.

**Figure 2 microorganisms-09-00217-f002:**
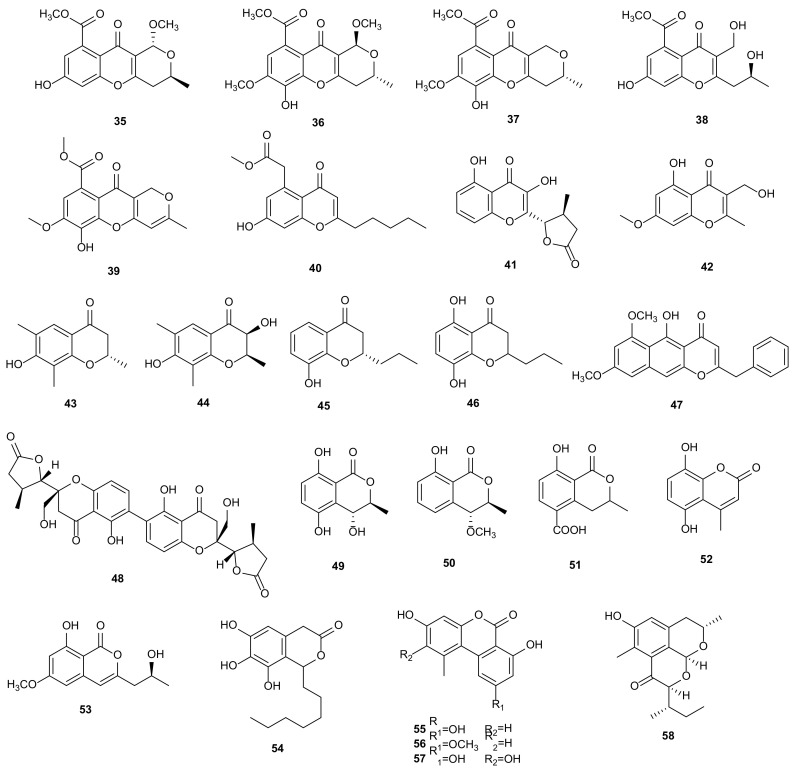
Chemical structures of compounds **35**–**58** from *Phomopsis*.

**Figure 3 microorganisms-09-00217-f003:**
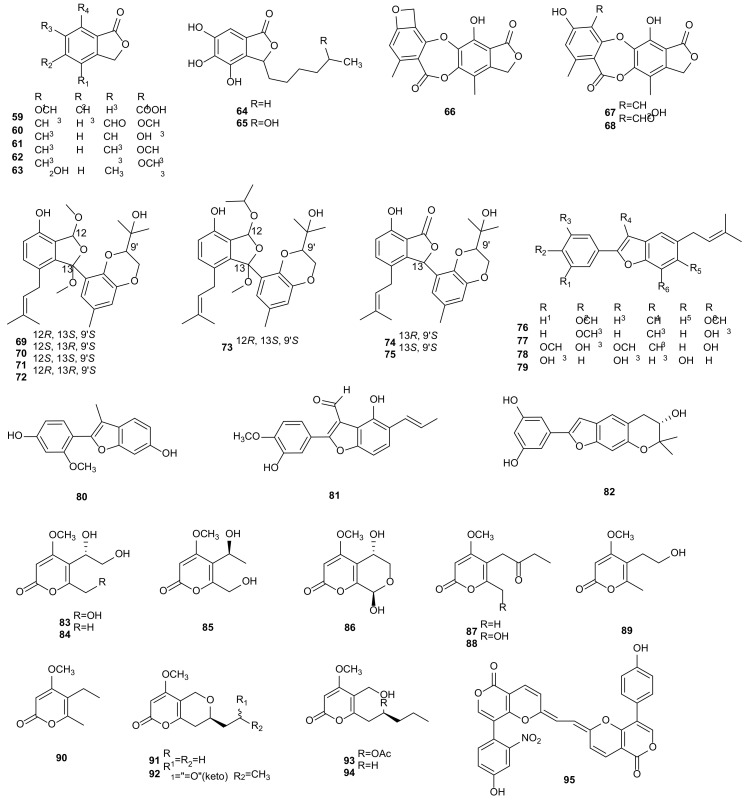
Chemical structures of compounds **59**–**95** from *Phomopsis*.

**Figure 4 microorganisms-09-00217-f004:**
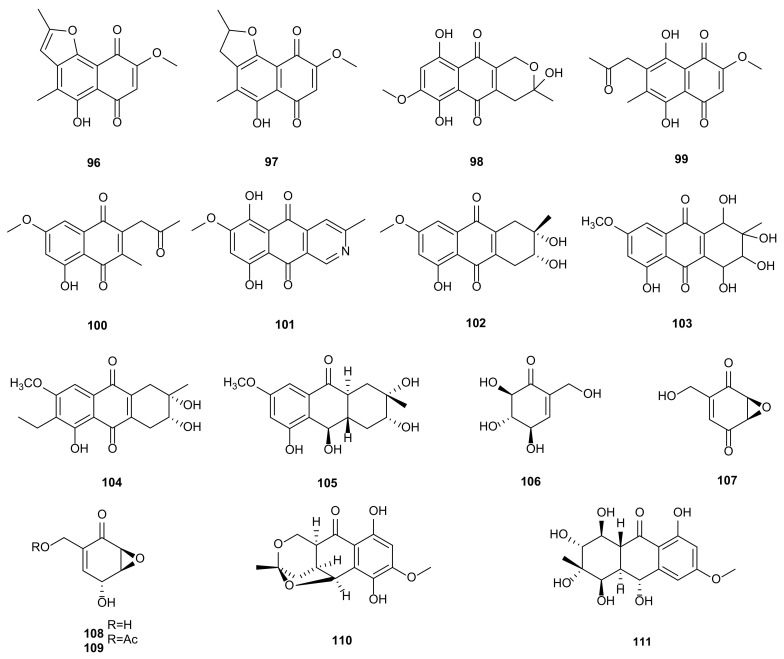
Chemical structures of compounds **96**–**111** from *Phomopsis*.

**Figure 5 microorganisms-09-00217-f005:**
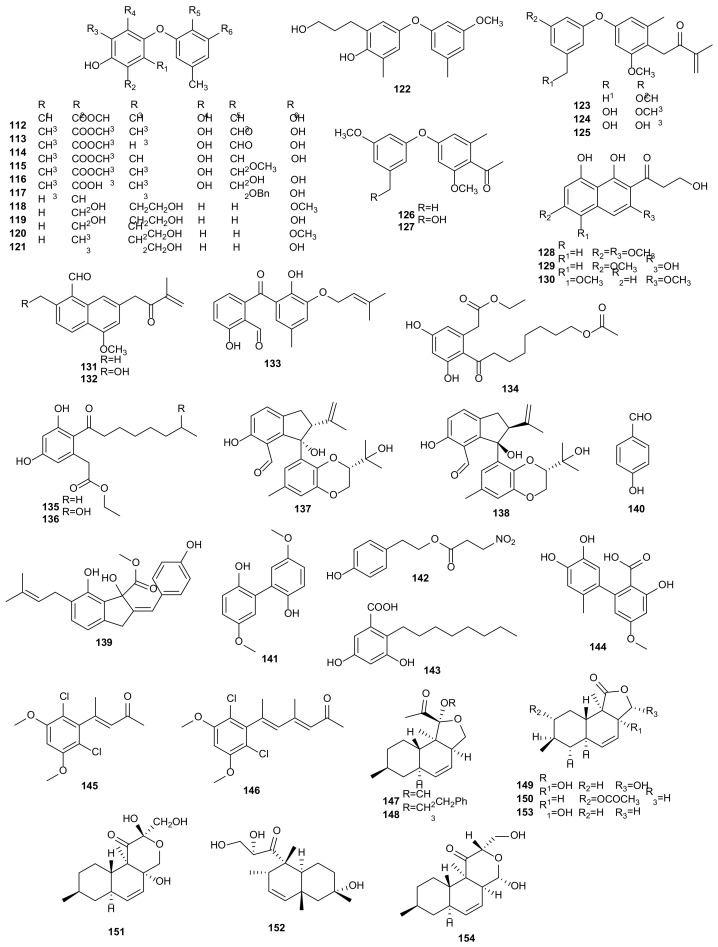
Chemical structures of compounds **112**–**154** from *Phomopsis*.

**Figure 6 microorganisms-09-00217-f006:**
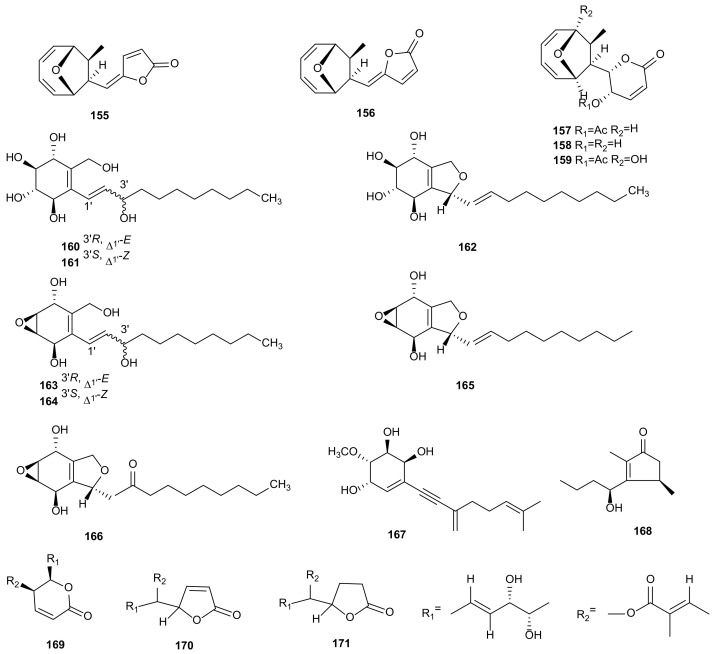
Chemical structures of compounds **155**–**171** from *Phomopsis*.

**Figure 7 microorganisms-09-00217-f007:**
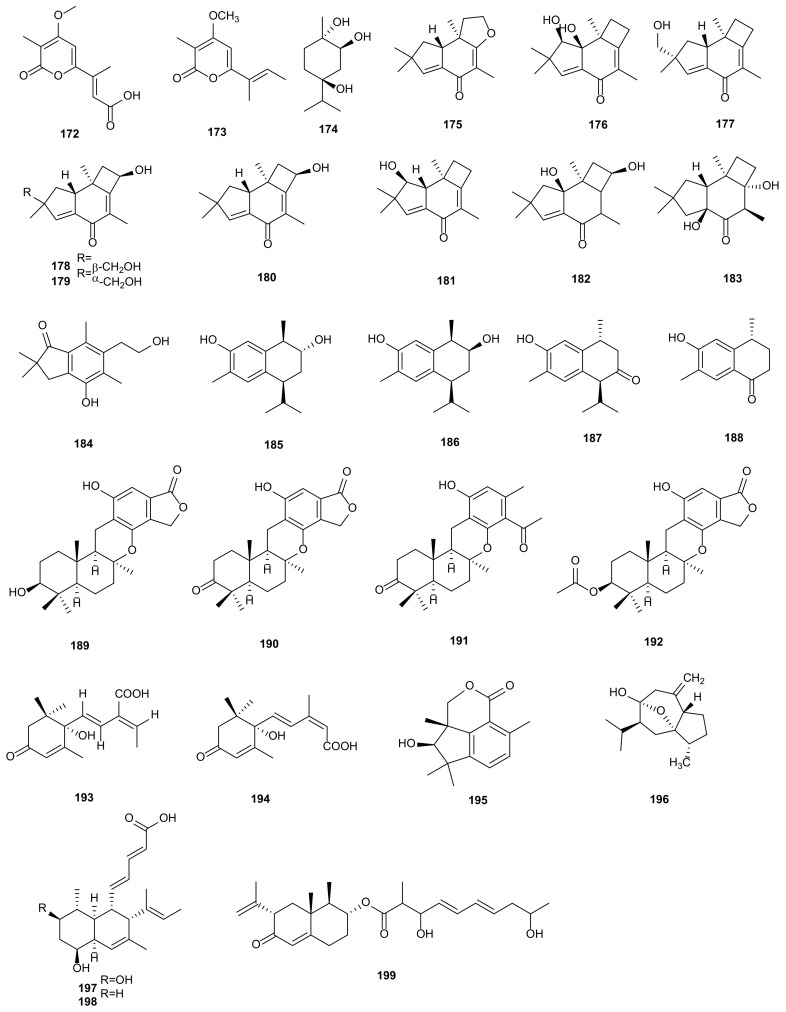
Chemical structures of compounds **172**–**199** from *Phomopsis*.

**Figure 8 microorganisms-09-00217-f008:**
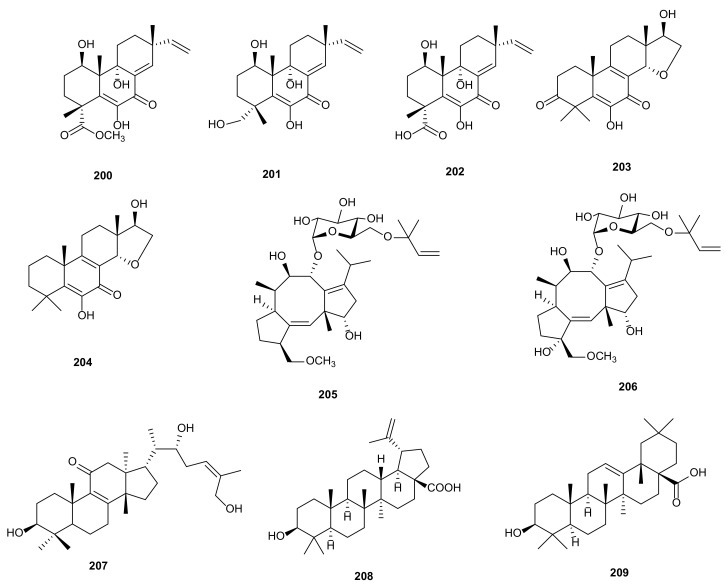
Chemical structures of compounds **200**–**209** from *Phomopsis*.

**Figure 9 microorganisms-09-00217-f009:**
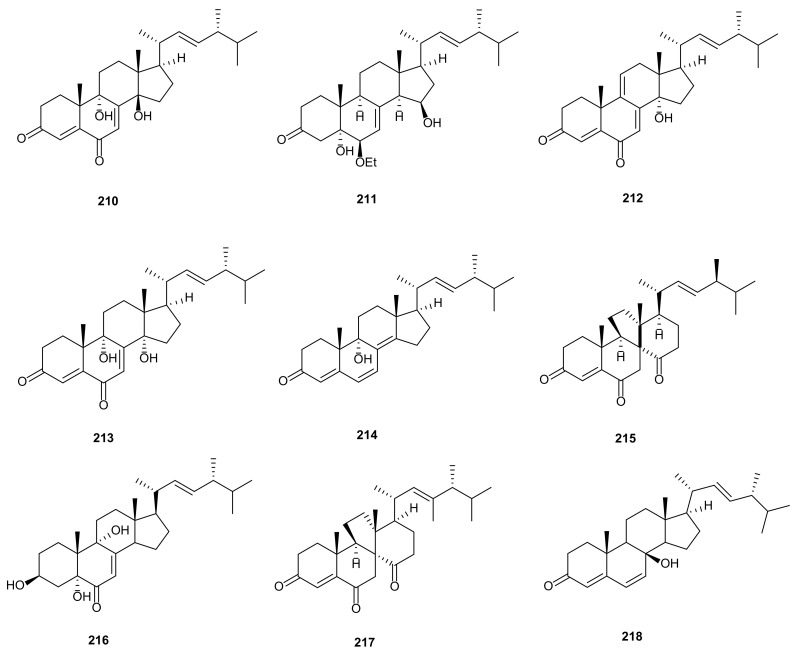
Chemical structures of compounds **210**–**218** from *Phomopsis*.

**Figure 10 microorganisms-09-00217-f010:**
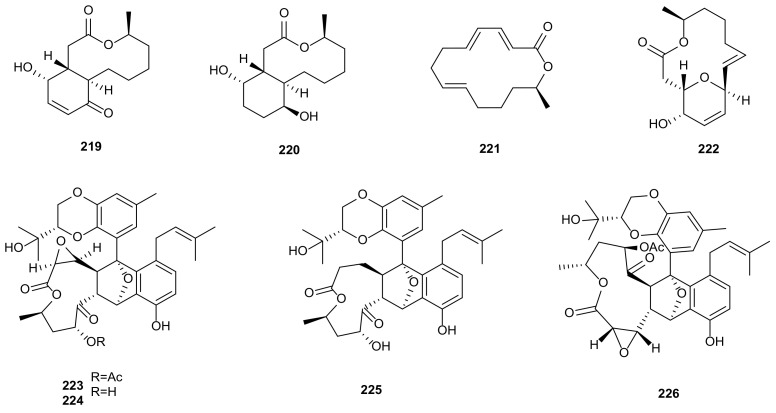
Chemical structures of compounds **219**–**226** from *Phomopsis*.

**Figure 11 microorganisms-09-00217-f011:**
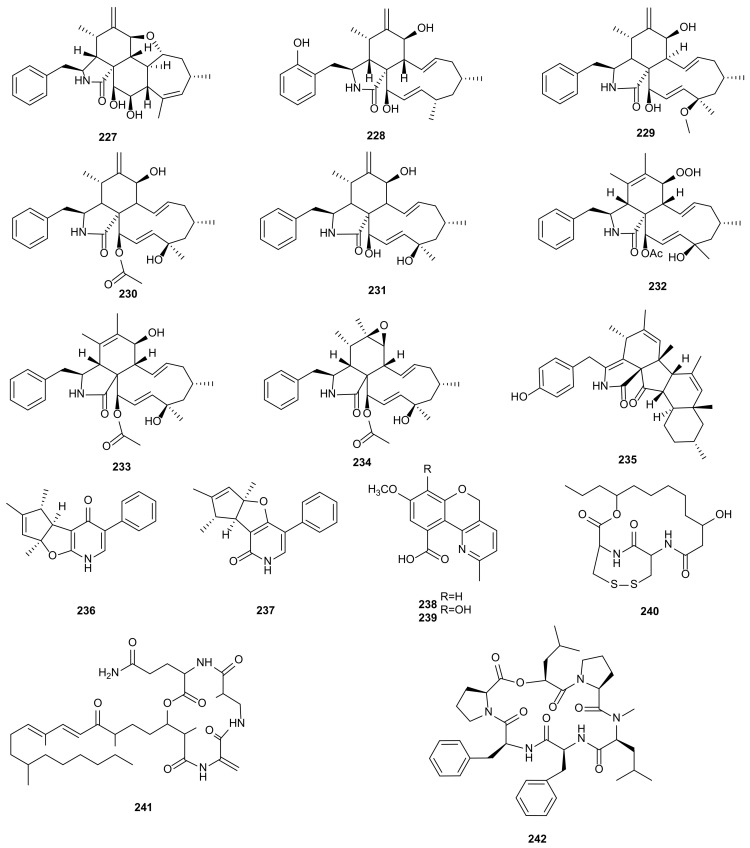
Chemical structures of compounds **227**–**242** from *Phomopsis*.

**Figure 12 microorganisms-09-00217-f012:**
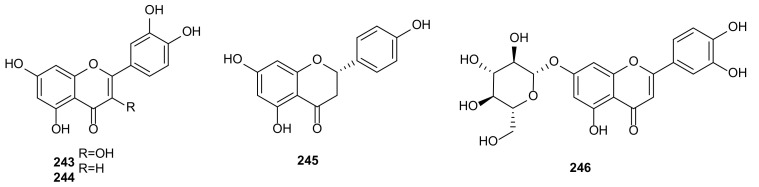
Chemical structures of compounds **243**–**246** from *Phomopsis*.

**Figure 13 microorganisms-09-00217-f013:**
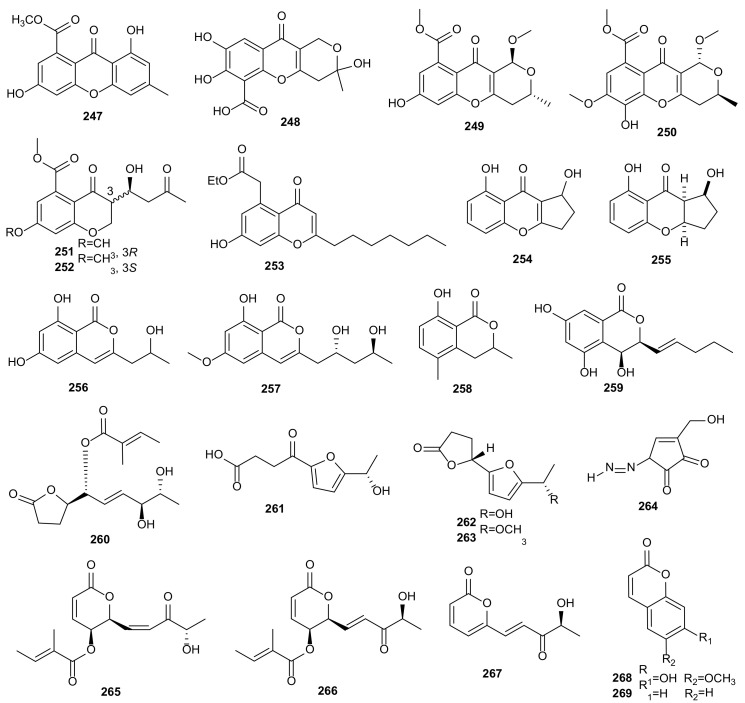
Chemical structures of compounds **247**–**269** from *Diaporthe*.

**Figure 14 microorganisms-09-00217-f014:**
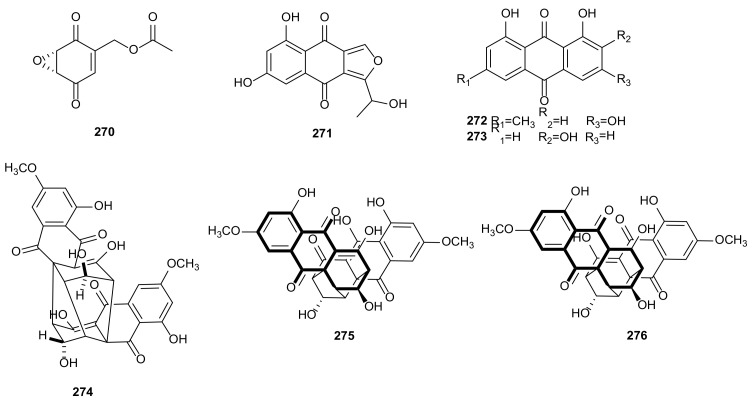
Chemical structures of compounds **270**–**276** from *Diaporthe*.

**Figure 15 microorganisms-09-00217-f015:**
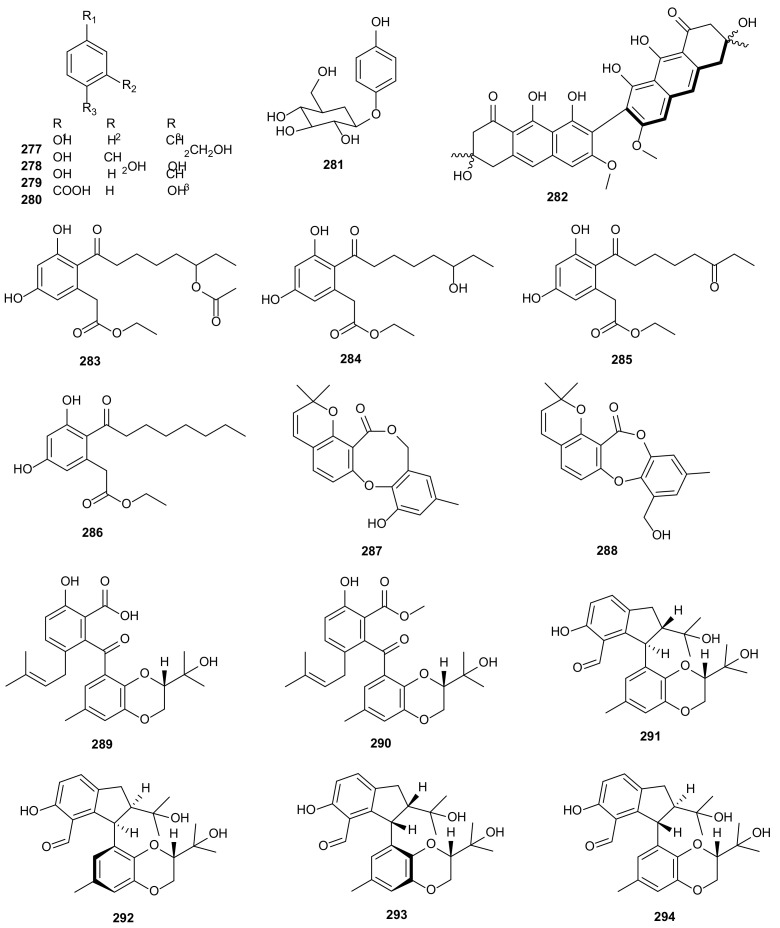
Chemical structures of compounds **277**–**294** from *Diaporthe*.

**Figure 16 microorganisms-09-00217-f016:**
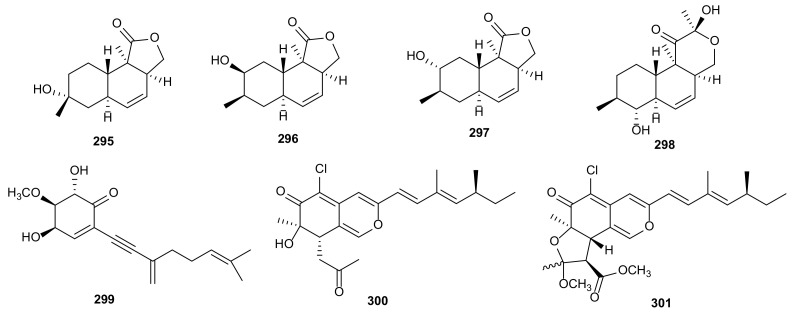
Chemical structures of compounds **295**–**301** from *Diaporthe*.

**Figure 17 microorganisms-09-00217-f017:**
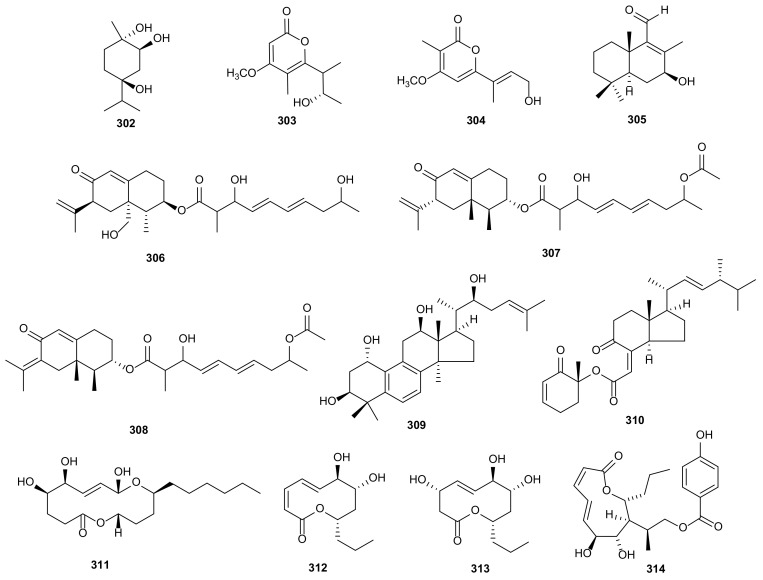
Chemical structures of compounds **302**–**314** from *Diaporthe*.

**Figure 18 microorganisms-09-00217-f018:**
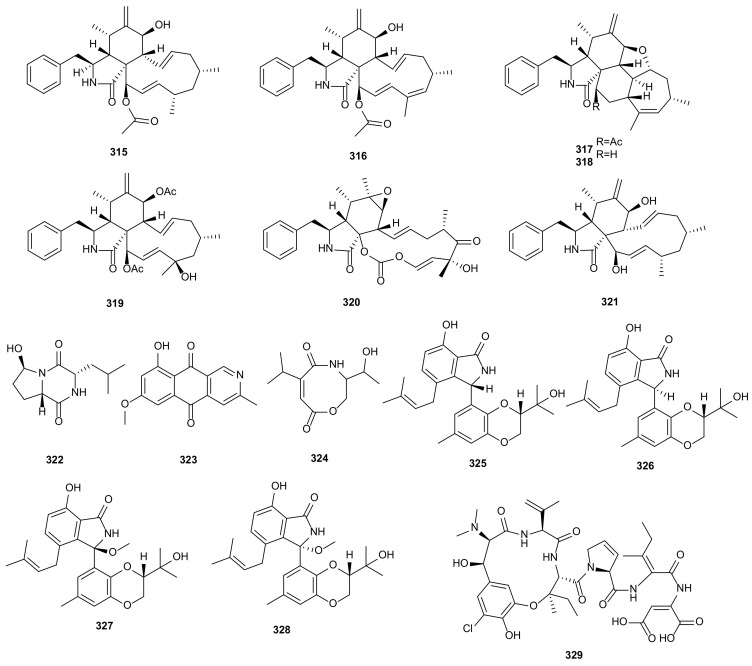
Chemical structures of compounds **315**–**329** from *Diaporthe*.

**Figure 19 microorganisms-09-00217-f019:**
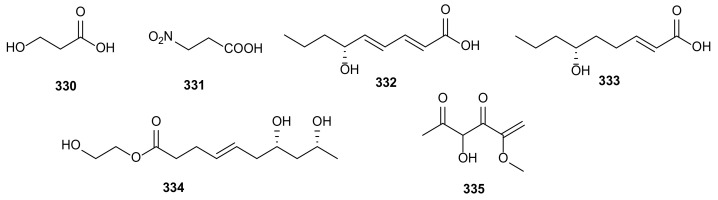
Chemical structures of compounds **330**–**335** from *Diaporthe*.

**Figure 20 microorganisms-09-00217-f020:**
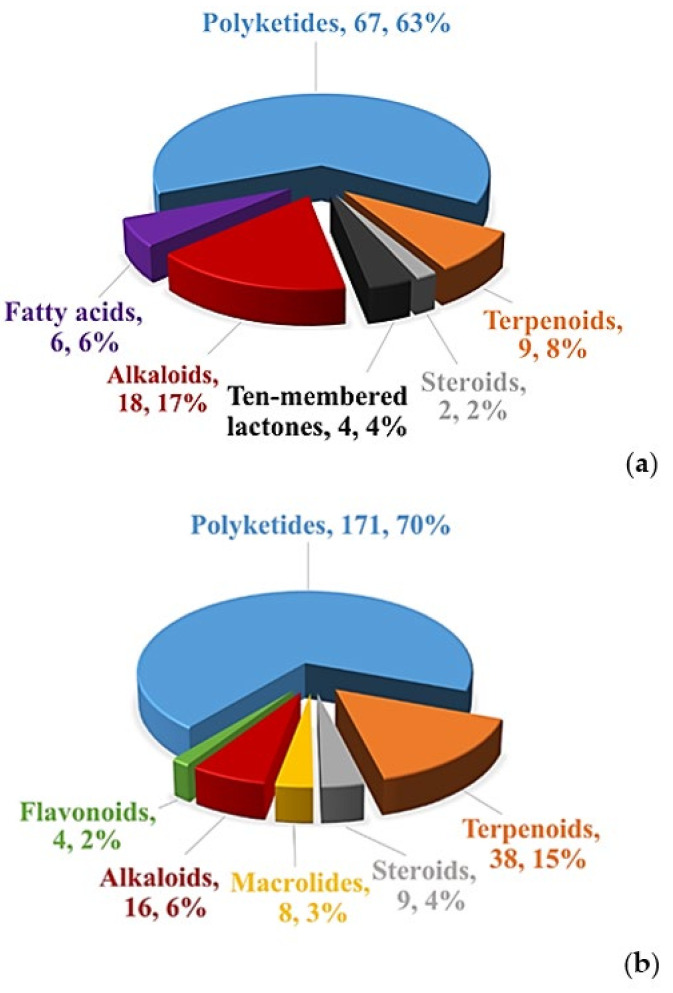
(**a**) The proportion of structural types of bioactive compounds from *Diaporthe*; (**b**) The proportion of structural types of bioactive compounds from *Phomopsis*.

**Figure 21 microorganisms-09-00217-f021:**
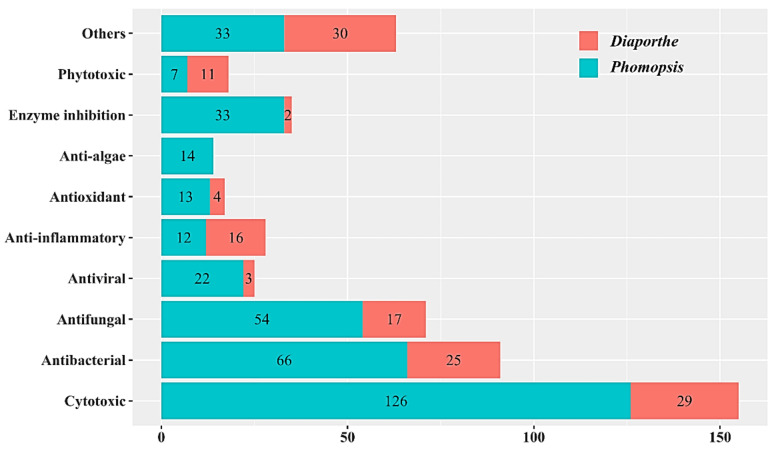
The distribution of main bioactivities of compounds isolated from *Diaporthe* and *Phomopsis*.

**Table 1 microorganisms-09-00217-t001:** The bioactive secondary metabolites of the anamorph *Phomopsis* during 2010–2019.

Number	Structural Types	Compounds	Strains	Habitats (T/M ^a^)	Activities	Refs.
1	Xanthones	1,5-Dihydroxy-3-hydroxyethyl-6-methoxy-carbonylxanthone	*Phomopsis* sp.	*Paris polyphylla* var*. yunnanensis* (T)	Cytotoxic	[23]
2		1-Hydroxy-5-methoxy-3-hydroxyethyl-6-methoxycarbonylxanthone	*Phomopsis* sp.	*P. polyphylla* var*. yunnanensis* (T)	Cytotoxic	[23]
3		1-Hydroxy-3-hydroxyethyl-8-ethoxycarbonyl-xanthone	*Phomopsis* sp.	*P. polyphylla* var*. yunnanensis* (T)	Cytotoxic	[23]
4		Pinselin	*Phomopsis* sp.	*P. polyphylla* var*. yunnanensis* (T)	Cytotoxic	[23]
5		1-Hydroxy-8-(hydroxymethyl)-3-methoxy-6-methylxanthone	*Phomopsis* sp.	*P. polyphylla* var*. yunnanensis* (T)	Cytotoxic	[23]
6		2,6-Dihydroxy-3-methyl-9-oxoxanthene-8-carboxylic acid methyl ester	*Phomopsis* sp. (No. SK7RN3G1)	Sediment (M)	Cytotoxic	[24]
7		4,5-Dihydroxy-3-(2-hydroxyethyl)-1-methoxy-8-methoxy- carbonylxanthone	*P. amygdali*	*Paris axialis* (T)	Cytotoxic	[25]
8		1,8-Dihydroxy-4-(2-hydroxyethyl)-3-methoxyxanthone	*P. amygdali*	*P*. *axialis* (T)	Cytotoxic	[25]
9		Hydroxyvertixanthone	*Phomopsis* sp. YM 355364	*Aconitum carmichaelii* (T)	Antimicrobial	[26]
10		Dalienxanthone A	*Phomopsis* sp.	*Paris daliensis* (T)	Cytotoxic	[27]
11		Dalienxanthone B	*Phomopsis* sp.	*P*. *daliensis* (T)	Cytotoxic	[27]
12		Dalienxanthone C	*Phomopsis* sp.	*P*. *daliensis* (T)	Cytotoxic	[27]
13		Paucinervin E	*P. amygdali*	*P*. *axialis* (T)	Cytotoxic	[25]
14		1,3-Dihydroxy-4-(1,3,4-trihydroxybutan-2-yl)-8-methoxy-9*H*-xanthen-9-one	*P. amygdali*	*P. polyphylla* var*. yunnanensis* (T)	Cytotoxic	[28]
15		3-Methoxy-1,4,8-trihydroxy-5-(1ʹ,3ʹ,4ʹ-trihydroxybutan-2ʹ-yl)-xanthone	*P. amygdali*	*P*. *axialis* (T)	Cytotoxic	[29]
16		8-Methoxy-1,3,4-trihydroxy-5-(1ʹ,3ʹ,4ʹ-trihydroxybutan-2ʹ-yl)-xanthone	*P. amygdali*	*P*. *axialis* (T)	Cytotoxic	[29]
17		Secosterigmatocystin	*Phomopsis* sp. *P. amygdali*	*P. polyphylla* var*. yunnanensis* (T) *P*. *axialis* (T)	Cytotoxic Cytotoxic	[23] [29]
18		3,8-Dihydroxy-4-(2,3-dihydroxy-1-hydroxymethylpropyl)-1-methoxyxanthone	*Phomopsis* sp.	*P*. *daliensis* (T)	Cytotoxic	[27]
19		Oliganthins E	*Phomopsis* sp.	*P*. *daliensis* (T)	Cytotoxic	[27]
20		Dihydrosterigmatocystin	*P. amygdali*	*P*. *axialis*(T)	Cytotoxic	[29]
21		Vieillardixanthone	*P. amygdali*	*P*. *axialis* (T)	Cytotoxic	[29]
22		1,7-Dihydroxy-2-methoxy-3-(3-methylbut-2-enyl)xanthone	*Phomopsis* sp.	*P. polyphylla* var*. yunnanensis* (T)	Cytotoxic	[23]
23		1-Hydroxy-4,7-dimethoxy-6-(3-oxobutyl)-xanthone	*Phomopsis* sp.	*P. polyphylla* var*. yunnanensis* (T)	Cytotoxic	[23]
24		Asperxanthone	*Phomopsis* sp.	*P. polyphylla* var. *yunnanensis* (T)	Cytotoxic	[23]
25		6-*O*-Methyl-2-deprenylrheediaxanthone B	*Phomopsis* sp.	*P. polyphylla* var*. yunnanensis* (T)	Cytotoxic	[23]
26		Cratoxylumxanthone D	*Phomopsis* sp.	*P*. *daliensis* (T)	Cytotoxic	[27]
27		3-*O*-(6-*O*-*α*-L-Arabinopyranosyl)-*β*-D-glucopyranosyl-1,4-dimethoxyxanthone	*Phomopsis* sp. (ZH76)	*Excoecaria agallocha* (M)	Cytotoxic	[30]
28		Phomoxanthone A	*P*. *longicolla* *Phomopsis* sp. IM 41-1 *Phomopsis* sp. 33#	*Sonneratia caseolaris* (M) *Rhizhopora mucronata* (M) *Rhizophora stylosa* (M)	Pro-apoptotic Antimicrobial Inhibiting acetylcholinesterase and *α*-glucosidase, Antioxidant	[31] [32] [33]
29		12-*O*-Deacetyl-phomoxanthone A	*Phomopsis* sp. IM 41-1	*R*. *mucronata* (M)	Antimicrobial	[32]
30		Dicerandrol A	*P*. *longicolla* S1B4 *Phomopsis* sp. HNY29-2B	- ^b^ *Acanthus ilicifolius* (M)	Antimicrobial Cytotoxic	[34] [35]
31		Dicerandrol B	*P*. *longicolla* S1B4 *Phomopsis* sp. HNY29-2B	- ^b^ *A*. *ilicifolius* (M)	Antibacterial Cytotoxic	[34] [35]
32		Dicerandrol C	*P*. *longicolla* S1B4	- ^b^	Antibacterial	[34]
33		Deacetylphomoxanthone B	*P*. *longicolla* S1B4 *Phomopsis* sp. HNY29-2B	- ^b^ *A*. *ilicifolius*(M)	Antibacterial Cytotoxic	[34] [35]
34		Penexanthone A	*Phomopsis* sp. HNY29-2B	*A*. *ilicifolius* (M)	Cytotoxic	[35]
35	Chromones	(+)-Phomopsichin A	*Phomopsis* sp. 33#	*R*. *stylosa* (M)	Antimicrobial, Antioxidant, Inhibiting acetylcholinesterase and *α*-glucosidase	[33]
36		(−)-Phomopsichin B	*Phomopsis* sp. 33#	*R*. *stylosa* (M)	Antimicrobial, Antioxidant, Inhibiting acetylcholinesterase and *α*-glucosidase	[33]
37		Phomopsichin C	*Phomopsis* sp. 33#	*R*. *stylosa* (M)	Antimicrobial, Antioxidant, Inhibiting acetylcholinesterase and *α*-glucosidase	[33]
38		Phomopsichin D	*Phomopsis* sp. 33#	*R*. *stylosa* (M)	Antimicrobial, Antioxidant, Inhibiting acetylcholinesterase and *α*-glucosidase	[33]
39		Chaetocyclinone B	*Phomopsis* sp. HNY29-2B	*A*. *ilicifolius* (M)	Cytotoxic	[36]
40		Pestalotiopsone F	*Phomopsis* sp. IFB-ZS1-S4	*Scaevola hainanensis* (M)	Inhibiting neuraminidase	[37]
41		Phomoxanthone F	*Phomopsis* sp. xy21	*Xylocarpus granatum* (M)	Anti-HIV	[38]
42		5-Hydroxy-3-hydroxymethyl-2-methyl-7-methoxychromone	*Phomopsis* sp. (No. Gx-4)	Sediment (M)	Cytotoxic, Inhibiting the growth of SIV branch	[39]
43		Phomochromone A	*Phomopsis* sp.	*Cistus monspeliensis* (T)	Antimicrobial, Antialgal	[40]
44		Phomochromone B	*Phomopsis* sp.	*C*. *monspeliensis* (T)	Antimicrobial, Antialgal	[40]
45		Phomochromanone A	*Phomopsis* sp. CGMCC No. 5416	*Achyranthes bidentata* (T)	Cytotoxic, Anti-HIV	[41]
46		Phomochromanone B	*Phomopsis* sp. CGMCC No. 5416	*A*. *bidentata* (T)	Cytotoxic, Anti-HIV	[41]
47		5-Hydroxy-6,8-dimethoxy-2-benzyl-4*H*-naphtho[2,3-b]-pyran-4-one	*Phomopsis* sp. ZSU-H26	*E*. *agallocha* (M)	Cytotoxic	[42]
48		Phomopsis-H76 A	*Phomopsis* sp. (#zsu-H76)	*E*. *agallocha* (M)	Accelerating the growth of SIV branch	[43]
49	Chromanones	(3*R*,4*S*)-3,4-Dihydro-4,5,8-trihydroxy-3-methylisocoumarin	*Phomopsis* sp. (No. ZH-111)	Sediment (M)	Accelerating the growth of SIV branch, Cytotoxic	[44]
50		(3*R*,4*S*)-3,4-Dihydro-8-hydroxy-4-methoxy-3-methylisocoumarin	*Phomopsis* sp. (No. Gx-4)	Sediment (M)	Cytotoxic, Accelerating the growth of SIV branch	[39]
51		3,4-Dihydro-8-hydroxy-3-methyl-1*H*-2-benzopyran-1-one-5-carboxylic acid	*Phomopsis* sp. (No. Gx-4)	Sediment (M)	Cytotoxic, Accelerating the growth of SIV branch	[39]
52		5,8-Dihydroxy-4-methylcoumarin	*Phomopsis* sp. (No. Gx-4)	Sediment (M)	Cytotoxic, Inhibiting the growth of SIV branch	[39]
53		(10*S*)-Diaporthin	*Phomopsis* sp. sh917	*Isodon eriocalyx* var*. laxiflora* (T)	Antiangiogenic	[45]
54		Cytosporone D	*Phomopsis* sp. CMU-LMA	*Alpinia malacensis* (T)	Antimicrobial, Inibiting DnaG primase	[46]
55		Alternariol	*Phomopsis* sp. A240 *Phomopsis* sp. CAFT69 *Phomopsis* sp.	*Taxus chinensis* var*. mairei* (T) *Endodesmia calophylloides* (T) *Senna spectabilis* (T)	Cytotoxic Motility inhibitory and zoosporicidal potential Anti-inflammatory	[47] [48] [49]
56		Alternariol-5-*O*-methyl ether	*Phomopsis* sp. CAFT69	*E*. *calophylloides* (T)	Motility inhibitory and zoosporicidal potential	[48]
57		5ʹ-Hydroxyalternariol	*Phomopsis* sp. A240 *Phomopsis* sp. CAFT69	*T. chinensis* var*. mairei* (T) *E*. *calophylloides* (T)	Antioxidant Motility inhibitory and zoosporicidal potential	[47] [48]
58		Phomochromanone C	*Phomopsis* sp. CGMCC No. 5416	*A*. *bidentata* (T)	Cytotoxic, Pro-apoptotic	[41]
59	Benzofuranones	7-Methoxy-6-methyl-3-oxo-1,3-dihydroisobenzofuran-4-carboxylic acid	*Phomopsis* sp. A123	*Kandelia candel* (M)	Cytotoxic, Antifungal, Antioxidant	[50]
60		Diaporthelactone	*Phomopsis* sp. A123	*K*. *candel* (M)	Cytotoxic, Antifungal, Antioxidant	[50]
61		7-Hydroxy-4,6-dimethy-3H-isobenzofuran-1-one	*Phomopsis* sp. A123	*K*. *candel* (M)	Cytotoxic, Antifungal, Antioxidant	[50]
62		7-Methoxy-4,6-dimethyl-3H-isobenzofuran-1-one	*Phomopsis* sp. A123	*K*. *candel* (M)	Cytotoxic, Antifungal, Antioxidant	[50]
63		4-(Hydroxymethyl)-7-methoxy-6-methyl-1(3*H*)-isobenzofuranone	*Phomopsis* sp. (No. ZH-111)	Sediment (M)	Inhibiting the growth of SIV branch, Cytotoxic	[44]
64		Cytosporone E	*Phomopsis* sp. BCC 45011	*X*. *granatum*(M)	Cytotoxic, Antimalarial	[51]
65		Cytosporone P	*Phomopsis* sp. BCC 45011	*X*. *granatum* (M)	Antimalarial	[51]
66		Phomopsidone A	*Phomopsis* sp. A123	*K*. *candel* (M)	Cytotoxic, Antifungal, Antioxidant	[50]
67		Excelsione	*Phomopsis* sp. A123	*K*. *candel* (M)	Cytotoxic, Antifungal, Antioxidant	[50]
68		Excelsional	*Phomopsis* sp. CAFT69	*E*. *calophylloides* (T)	Motility inhibitory and zoosporicidal potential	[48]
69		Lithocarol A	*P*. *lithocarpus* FS508	Sediment (M)	Cytotoxic	[52]
70		Lithocarol B	*P*. *lithocarpus* FS508	Sediment (M)	Cytotoxic	[52]
71		Lithocarol C	*P*. *lithocarpus* FS508	Sediment (M)	Cytotoxic	[52]
72		Lithocarol D	*P*. *lithocarpus* FS508	Sediment (M)	Cytotoxic	[52]
73		Lithocarol E	*P*. *lithocarpus* FS508	Sediment (M)	Cytotoxic	[52]
74		Lithocarol F	*P*. *lithocarpus* FS508	Sediment (M)	Cytotoxic	[52]
75		Isoprenylisobenzofuran A	*P*. *lithocarpus* FS508	Sediment (M)	Cytotoxic	[52]
76		7-Methoxy-2-(4-methoxyphenyl)-3-methyl-5-(3-prenyl)-benzofuran	*Phomopsis* sp.	*P. polyphylla* var*. yunnanensis* (T)	Anti-TMV	[53]
77		2-(4-Methoxyphenyl)-3-methyl-5-(3-prenyl)-benzofuran-7-ol	*Phomopsis* sp.	*P. polyphylla* var*. yunnanensis* (T)	Anti-TMV	[53]
78		2-(4-Hydroxy-3,5-dimethoxyphenyl)-3-methyl-5-(3-prenyl) benzofuran-7-ol	*Phomopsis* sp.	*P. polyphylla* var*. yunnanensis* (T)	Anti-TMV	[53]
79		Moracin N	*Phomopsis* sp.	*P. polyphylla* var*. yunnanensis* (T)	Anti-TMV	[53]
80		2-(2′-Methoxy-4′-hydroxy)-aryl-3-methy-6-hydroxybenzofuran	*Phomopsis* sp.	*P. polyphylla* var*. yunnanensis* (T)	Anti-TMV	[53]
81		Iteafuranal B	*Phomopsis* sp.	*P. polyphylla* var*. yunnanensis* (T)	Anti-TMV	[53]
82		Moracin P	*Phomopsis* sp.	*P. polyphylla* var*. yunnanensis* (T)	Anti-TMV	[53]
83	Pyrones	Phomaspyrone A	*P*. *asparagi* SWUKJ5.2020	*Kadsura angustifolia* (T)	Cytotoxic	[54]
84		Macommelin-8,9-diol	*P*. *asparagi* SWUKJ5.2020	*K*. *angustifolia* (T)	Cytotoxic	[54]
85		Phomaspyrone B	*P*. *asparagi* SWUKJ5.2020	*K*. *angustifolia* (T)	Cytotoxic	[54]
86		Phomaspyrone C	*P*. *asparagi* SWUKJ5.2020	*K*. *angustifolia* (T)	Cytotoxic	[54]
87		Phomaspyrone D	*P*. *asparagi* SWUKJ5.2020	*K*. *angustifolia* (T)	Cytotoxic	[54]
88		Phomaspyrone E	*P*. *asparagi* SWUKJ5.2020	*K*. *angustifolia* (T)	Cytotoxic	[54]
89		Macommelin-9-ol	*P*. *asparagi* SWUKJ5.2020	*K*. *angustifolia* (T)	Cytotoxic	[54]
90		Macommelin	*P*. *asparagi* SWUKJ5.2020	*K*. *angustifolia* (T)	Cytotoxic	[54]
91		Pyrenocine J	*Phomopsis* sp.	*Cistus salvifolius* (T)	Antifungal, Antibacterial, Algicidal	[55]
92		Pyrenocine K	*Phomopsis* sp.	*C*. *salvifolius* (T)	Antifungal, Antibacterial, Algicidal	[55]
93		Pyrenocine L	*Phomopsis* sp.	*C*. *salvifolius* (T)	Antibacterial, Algicidal	[55]
94		Pyrenocine M	*Phomopsis* sp.	*C*. *salvifolius* (T)	Antifungal, Antibacterial, Algicidal	[55]
95		Phomopsis-H76 C	*Phomopsis* sp. (#zsu-H76)	*E*. *agallocha* (M)	Inhibiting the growth of SIV branch	[43]
96	Quinones	Anhydrojavanicin	*Phomopsis* sp. HCCB04730	*Radix Stephaniae Japonicae* (T)	Cytotoxic, Anti-HIV	[56]
97		Dihydroanhydrojavanicin	*Phomopsis* sp. HCCB04730	*Radix Stephaniae Japonicae* (T)	Cytotoxic, Anti-HIV	[56]
98		Fusarubin	*Phomopsis* sp. HCCB04730	*Radix Stephaniae Japonicae* (T)	Cytotoxic, Anti-HIV	[56]
99		Javanicin	*Phomopsis* sp. HCCB04730	*Radix Stephaniae Japonicae* (T)	Cytotoxic, Anti-HIV	[56]
100		2-Acetonyl-3methyl-5-hydroxy-7-methoxy-naphthazarin	*Phomopsis* sp. HCCB04730	*Radix Stephaniae Japonicae* (T)	Cytotoxic, Anti-HIV	[56]
101		Bostrycoidin	*Phomopsis* sp. HCCB04730	*Radix Stephaniae Japonicae* (T)	Cytotoxic, Anti-HIV	[56]
102		Altersolanol B	*P*. *longicolla* HL-2232	*Bruguiera sexangula* var*. rhynchopetala* (M)	Antibacterial	[57]
103		Altersolanol A	*Phomopsis* sp. (PM0409092) *P*. *foeniculi*	*Nyctanthes arbor-tristis* (T) *Foeniculum vulgare* (T)	Cytotoxic Phytotoxic	[58] [59]
104		(2*R*,3*S*)-7-Ethyl-1,2,3,4-tetrahydro-2,3,8-trihdroxy-6-methoxy-3-methyl-9,10-anthracenedione	*Phomopsis* sp. PSU-MA214	*Rhizophora apiculata* (M)	Cytotoxic, Antibacterial	[60]
105		Altersolanol J	*P*. *foeniculi*	*F*. *vulgare* (T)	Phytotoxic	[59]
106		2-Hydroxymethyl-4*β*,5*α*,6*β*-trihydroxycyclohex-2-en	*Phomopsis* sp.	*Notobasis syriaca* (T)	Antibacterial, Algicidal	[61]
107		(−)-Phyllostine	*Phomopsis* sp.	*N*. *syriaca* (T)	Antifungal, Antibacterial, Algicidal	[61]
108		(+)-Epiepoxydon	*Phomopsis* sp.	*N*. *syriaca* (T)	Antibacterial, Algicidal	[61]
109		(+)-Epoxydon monoacetate	*Phomopsis* sp.	*N*. *syriaca* (T)	Antifungal, Antibacterial, Algicidal	[61]
110		Phomonaphthalenone A	*Phomopsis* sp. HCCB04730	*Radix Stephaniae Japonicae* (T)	Cytotoxic, Anti-HIV	[56]
111		Ampelanol	*Phomopsis* sp. HNY29-2B	*A*. *ilicifolius* (M)	Antibacterial	[62]
112	Phenols	Phomosine K	*Phomopsis* sp.	*N*. *syriaca* (T)	Antibacterial	[61]
113		Phomosine A	*Phomopsis* sp.	*Ligustrum vulgare* (T)	Antifungal, Antibacterial, Inhibiting algae	[63]
114		Phomosine B	*Phomopsis* sp.	*L*. *vulgare* (T)	Antifungal, Antibacterial	[63]
115		Phomosine C	*Phomopsis* sp.	*L*. *vulgare* (T)	Antifungal, Antibacterial	[63]
116		Phomosine D	*Phomopsis* sp.	*L*. *vulgare* (T)	Antifungal, Inhibiting algae	[63]
117		Phomosine I	*Phomopsis* sp.	*L*. *vulgare* (T)	Antifungal, Antibacterial	[63]
118		4-(3-Methoxy-5-methylphenoxy)-2-(2-hydroxyethyl)-6-(hydroxymethyl)phenol	*P*. *asparagi*	*P. polyphylla* var*. yunnanensis* (T)	Anti-MRSA	[64]
119		4-(3-Hydroxy-5-methylphenoxy)-2-(2-hydroxyethyl)-6-(hydroxymethyl)phenol	*P*. *asparagi*	*P. polyphylla* var*. yunnanensis* (T)	Anti-MRSA	[64]
120		4-(3-Methoxy-5-methylphenoxy)-2-(2-hydroxyethyl)-6-methylphenol	*P*. *fukushii*	*P. polyphylla* var*. yunnanensis* (T)	Anti-MRSA	[65]
121		4-(3-Hydroxy-5-methylphenoxy)-2-(2-hydroxyethyl)-6-methylphenol	*P*. *fukushii*	*P. polyphylla* var*. yunnanensis* (T)	Anti-MRSA	[65]
122		4-(3-Methoxy-5-methylphenoxy)-2-(3-hydroxypropyl)-6-methylphenol	*P*. *fukushii*	*P. polyphylla* var*. yunnanensis* (T)	Anti-MRSA	[65]
123		1-(4-(3-Methoxy-5-methylphenoxy)-2-methoxy-6-methylphenyl)-3-methylbut-3-en-2-one	*P*. *fukushii*	*P. polyphylla* var*. yunnanensis* (T)	Anti-MRSA	[66]
124		1-(4-(3-(Hydroxymethyl)-5methoxyphenoxy)-2-methoxy-6-methylphenyl)-3-methylbut-3-en-2-one	*P*. *fukushii*	*P. polyphylla* var*. yunnanensis* (T)	Anti-MRSA	[66]
125		1-(4-(3-Hydroxy-5(hydroxymethyl)phenoxy)-2-methoxy-6-methylphenyl)-3-methylbut-3-en-2-one	*P*. *fukushii*	*P. polyphylla* var*. yunnanensis* (T)	Anti-MRSA	[66]
126		1-[2-Methoxy-4-(3-methoxy-5-methylphenoxy)-6-methylphenyl]-ethanone	*P*. *fukushii*	*P. polyphylla* var*. yunnanensis* (T)	Anti-MRSA	[67]
127		1-[4-(3-(Hydroxymethyl)-5-methoxyphenoxy)-2-methoxy-6-methylphenyl]-ethanone	*P*. *fukushii*	*P. polyphylla* var*. yunnanensis* (T)	Anti-MRSA	[67]
128		3-Hydroxy-1-(1,8-dihydroxy-3,6-dimethoxynaphthalen-2-yl)propan-1-one	*P*. *fukushii*	*P. polyphylla* var*. yunnanensis* (T)	Anti-MRSA	[68]
129		3-Hydroxy-1-(1,3,8-trihydroxy-6-methoxynaphthalen-2-yl)propan-1-one	*P*. *fukushii*	*P. polyphylla* var*. yunnanensis* (T)	Anti-MRSA	[68]
130		3-Hydroxy-1-(1,8-dihydroxy-3,5-dimethoxynaphthalen-2-yl)propan-1-one	*P*. *fukushii*	*P. polyphylla* var*. yunnanensis* (T)	Anti-MRSA	[68]
131		5-Methoxy-2-methyl-7-(3-methyl-2-oxobut-3-enyl)-1-naphthaldehyde	*Phomopsis* sp.	*P. polyphylla* var*. yunnanensis* (T)	Anti-MRSA	[69]
132		2-(Hydroxymethyl)-5-methoxy-7-(3-methyl-2-oxobut-3-enyl)-1-naphthaldehyde	*Phomopsis* sp.	*P. polyphylla* var*. yunnanensis* (T)	Anti-MRSA	[69]
133		Tenellone H	*P*. *lithocarpus* FS508	Sediment (M)	Cytotoxic	[70]
134		16-Acetoxycytosporone B	*Phomopsis* sp. YM 355364	*A*. *carmichaeli* (T)	Antifungal	[71]
135		Cytosporone B	*Phomopsis* sp. 0391 *Phomopsis* sp. PSU-H188	*P. polyphylla* var*. yunnanensis* (T) *Hevea brasiliensis* (T)	Inhibiting lipase Protecting pancreatic *β*-cells	[72] [73]
136		Dothiorelone A	*Phomopsis* sp. 0391	*P. polyphylla* var*. yunnanensis* (T)	Inhibiting lipase	[72]
137		Lithocarpinol A	*P*. *lithocarpus* FS508	Sediment (M)	Cytotoxic	[74]
138		Lithocarpinol B	*P*. *lithocarpus* FS508	Sediment (M)	Cytotoxic	[74]
139		Phomoindene A	*Phomopsis* sp. (No. GX7-4A)	Sediment (M)	Cytotoxic	[75]
140		4-Hydroxybenzaldehyde	*Phomopsis* sp. YM 355364	*A*. *carmichaelii* (T)	Antimicrobial	[26]
141		5,5′-Dimethoxybiphenyl-2,2′-diol	*P*. *longicolla* HL-2232	*B. sexangula* var*. rhynchopetala* (M)	Antibacterial	[57]
142		Phomonitroester	*Phomopsis* sp. PSU-MA214	*R*. *apiculate* (M)	Cytotoxic	[60]
143		Cytosporone U	*Phomopsis* sp. FJBR-11	*Brucea javanica* (T)	Anti-TMV	[76]
144		Altenusin	*Phomopsis* sp. CAFT69	*E*. *calophylloides* (T)	Motility inhibitory and zoosporicidal potential	[48]
145		Cosmochlorin D	*Phomopsis* sp. N-125	*Ficus ampelas* (T)	Cytotoxic, Growth-inhibition activity	[77]
146		Cosmochlorin E	*Phomopsis* sp. N-125	*F*. *ampelas* (T)	Cytotoxic, Growth-inhibition activity	[77]
147	Oblongolides	Oblongolide Z	*Phomopsis* sp. BCC 9789	*Musa acuminate* (T)	Cytotoxic, Anti-HSV-1	[78]
148		Oblongolide Y	*Phomopsis* sp. BCC 9789	*M*. *acuminate* (T)	Cytotoxic	[78]
149		Oblongolide C1	*Phomopsis* sp. XZ-01	*Camptotheca acuminate* (T)	Cytotoxic	[79]
150		Oblongolide P1	*Phomopsis* sp. XZ-01	*C*. *acuminate* (T)	Cytotoxic	[79]
151		Oblongolide X1	*Phomopsis* sp. XZ-01	*C*. *acuminate* (T)	Cytotoxic	[79]
152		6-Hydroxyphomodiol	*Phomopsis* sp. XZ-01	*C*. *acuminate* (T)	Cytotoxic	[79]
153		Oblongolide C	*Phomopsis* sp. XZ-01	*C*. *acuminate* (T)	Cytotoxic	[79]
154		2-Deoxy-4*α*-hydroxyoblongolide X	*Phomopsis* sp. BCC 9789	*M*. *acuminate* (T)	Anti-HSV-1	[78]
155	Unclassified polyketides	Phomoxydiene C	*Phomopsis* sp. BCC 45011	*X*. *granatum* (M)	Cytotoxic, Antimalarial	[51]
156		1893 A	*Phomopsis* sp. BCC 45011	*X*. *granatum* (M)	Cytotoxic	[51]
157		Mycoepoxydiene	*Phomopsis* sp. BCC 45011	*X*. *granatum* (M)	Cytotoxic, Antimalarial	[51]
158		Deacetylmycoepoxydiene	*Phomopsis* sp. BCC 45011	*X*. *granatum* (M)	Cytotoxic, Antimalarial	[51]
159		Phomoxydiene A	*Phomopsis* sp. BCC 45011	*X*. *granatum* (M)	Cytotoxic, Antimalarial	[51]
160		Phomopoxide A	*Phomopsis* sp. YE3250	*Paeonia delavayi* (T)	Cytotoxic, Antifungal, Inhibiting *α*-glycosidase	[80]
161		Phomopoxide B	*Phomopsis* sp. YE3250	*P*. *delavayi* (T)	Cytotoxic, Antifungal, Inhibiting *α*-glycosidase	[80]
162		Phomopoxide C	*Phomopsis* sp. YE3250	*P*. *delavayi* (T)	Cytotoxic, Antifungal, Inhibiting *α*-glycosidase	[80]
163		Phomopoxide D	*Phomopsis* sp. YE3250	*P*. *delavayi* (T)	Cytotoxic, Antifungal, Inhibiting *α*-glycosidase	[80]
164		Phomopoxide E	*Phomopsis* sp. YE3250	*P*. *delavayi* (T)	Cytotoxic, Antifungal, Inhibiting *α*-glycosidase	[80]
165		Phomopoxide F	*Phomopsis* sp. YE3250	*P*. *delavayi* (T)	Cytotoxic, Antifungal, Inhibiting *α*-glycosidase	[80]
166		Phomopoxide G	*Phomopsis* sp. YE3250	*P*. *delavayi* (T)	Cytotoxic, Antifungal, Inhibiting *α*-glycosidase	[80]
167		Phomentrioloxin	*Phomopsis* sp.	*Carthamus lanatus* (T)	Phytotoxic	[81]
168		Phomotenone	*Phomopsis* sp.	*C*. *monspeliensis* (T)	Antifungal, Antibacterial, Antialgal	[40]
169		Phomopsolide B	*Phomopsis* sp. DC275	*Vitis vinifera* (T)	Antibacterial, Phytotoxic	[82]
170		Phomopsolidone A	*Phomopsis* sp. DC275	*V*. *vinifera* (T)	Antibacterial, Phytotoxic	[82]
171		Phomopsolidone B	*Phomopsis* sp. DC275	*V*. *vinifera* (T)	Antibacterial, Phytotoxic	[82]
172	Monoterpenoids	Acropyrone	*Phomopsis* sp. HNY29-2B	*A*. *ilicifolius* (M)	Antibacterial	[62]
173		Nectriapyrone	*P*. *foeniculi*	*F*. *vulgare* (T)	Phytotoxic	[59]
174		(1*S*,2*S*,4*S*)-Trihydroxy-*p*-menthane	*Phomopsis* sp.	*C*. *monspeliensis* (T)	Antibacterial, Antialgal	[40]
175	Sesquiterpenoids	Phomophyllin A	*Phomopsis* sp. TJ507A	*Phyllanthus glaucus* (T)	Inhibiting BACE1	[83]
176		Phomophyllin B	*Phomopsis* sp. TJ507A	*P*. *glaucus* (T)	Inhibiting BACE1	[83]
177		Phomophyllin C	*Phomopsis* sp. TJ507A	*P*. *glaucus* (T)	Inhibiting BACE1	[83]
178		Phomophyllin D	*Phomopsis* sp. TJ507A	*P*. *glaucus* (T)	Inhibiting BACE1	[83]
179		Phomophyllin E	*Phomopsis* sp. TJ507A	*P*. *glaucus* (T)	Inhibiting BACE1	[83]
180		Phomophyllin F	*Phomopsis* sp. TJ507A	*P*. *glaucus* (T)	Inhibiting BACE1	[83]
181		Phomophyllin G	*Phomopsis* sp. TJ507A	*P*. *glaucus* (T)	Inhibiting BACE1	[83]
182		Radulone B	*Phomopsis* sp. TJ507A	*P*. *glaucus* (T)	Inhibiting BACE1	[83]
183		Phomophyllin I	*Phomopsis* sp. TJ507A	*P*. *glaucus* (T)	Inhibiting BACE1	[83]
184		Onitin	*Phomopsis* sp. TJ507A	*P*. *glaucus* (T)	Inhibiting BACE1	[83]
185		(7*R*,9*S*,10*R*)-3,9-Di-hidroxicalamenene	*P*. *cassiae*	*Cassia spectabilis* (T)	Inhibiting acetylcholinesterase, Antifungal	[84]
186		(7*R*,9*R*,10*R*)-3,9-Di-hidroxicalamenene	*P*. *cassiae*	*C*. *spectabilis* (T)	Inhibiting acetylcholinesterase, Antifungal	[84]
187		(7*S*,10*R*)-3-Hidroxicalamen-8-one	*P*. *cassiae*	*C*. *spectabilis* (T)	Inhibiting acetylcholinesterase, Antifungal	[84]
188		Aristelegone-A	*P*. *cassiae*	*C*. *spectabilis* (T)	Inhibiting acetylcholinesterase, Antifungal	[84]
189		Phomoarcherin A	*P*. *archeri*	*Vanilla albidia* (T)	Cytotoxic	[85]
190		Phomoarcherin B	*P*. *archeri*	*V*. *albidia* (T)	Cytotoxic, Antimalarial	[85]
191		Phomoarcherin C	*P*. *archeri*	*V*. *albidia* (T)	Cytotoxic	[85]
192		Kampanol A	*P*. *archeri*	*V*. *albidia* (T)	Cytotoxic	[85]
193		(+)-*S*-1-Methyl-abscisic-6-acid	*P*. *amygdali*	*Call midge* (T)	Antibacterial	[86]
194		(+)-*S*-Abscisic acid	*P*. *amygdali*	*C*. *midge* (T)	Antibacterial	[86]
195		7-Hydroxy-10-oxodehydrodihydrobotrydial	*Phomopsis* sp. TJ507A	*P*. *glaucus* (T)	Inhibiting BACE1	[83]
196		Curcumol	*P*. *castaneae-mollissimae* GQH87	*Artemisia annua* (T)	Cytotoxic	[87]
197		9-Hydroxyphomopsidin	*Phomopsis* sp. CAFT69	*E*. *calophylloides* (T)	Motility inhibitory and zoosporicidal potential	[48]
198		Phomopsidin	*Phomopsis* sp. CAFT69	*E*. *calophylloides* (T)	Motility inhibitory and zoosporicidal potential	[48]
199		AA03390	*P*. *lithocarpus* FS508	Sediment (M)	Cytotoxic	[70]
200	Diterpenoids	Libertellenone J	*Phomopsis* sp. S12	*Illigera rhodantha* (T)	Anti-inflammatory	[88]
201		Libertellenone C	*Phomopsis* sp. S12	- ^b^	Anti-inflammatory	[89]
202		Libertellenone T	*Phomopsis* sp. S12	- ^b^	Anti-inflammatory	[89]
203		Pedinophyllol K	*Phomopsis* sp. S12	- ^b^	Anti-inflammatory	[89]
204		Pedinophyllol L	*Phomopsis* sp. S12	- ^b^	Anti-inflammatory	[89]
205		Fusicoccin J	*P*. *amygdali*	*C*. *midge* (T)	Antibacterial	[86]
206		3*α*-Hydroxyfusicoccin J	*P*. *amygdali*	*C*. *midge* (T)	Antibacterial	[86]
207	Triterpenoids	3*S*,22*R*,26-Trihydroxy-8,24*E*-euphadien-11-one	*P*. *chimonanthi*	*Tamarix chinensis* (T)	Cytotoxic	[90]
208		Betulinic acid	*Phomopsis* sp. SNB-LAP1-7-32	*Diospyros carbonaria* (T)	Antiviral, Cytotoxic	[91]
209		Oleanolic acid	*P*. *castaneae-mollissi* *mae* GQH87	*A*. *annua* (T)	Cytotoxic	[87]
210	Steroids	(14*β*,22*E*)-9,14-Dihydroxyergosta-4,7,22-triene-3,6-dione	*Phomopsis* sp.	*A*. *carmichaeli* (T)	Antifungal	[92]
211		(5*α*,6*β*,15*β*,22*E*)-6-Ethoxy-5,15-dihydroxyergosta-7,22-dien-3-one	*Phomopsis* sp.	*A*. *carmichaeli* (T)	Antifungal	[92]
212		Calvasterol A	*Phomopsis* sp.	*A*. *carmichaeli* (T)	Antifungal	[92]
213		Calvasterol B	*Phomopsis* sp.	*A*. *carmichaeli* (T)	Antifungal	[92]
214		Ganodermaside D	*Phomopsis* sp.	*A*. *carmichaeli* (T)	Antifungal	[92]
215		Dankasterone A	*Phomopsis* sp. YM 355364	*A*. *carmichaeli* (T)	Antifungal, Anti-influenza	[71]
216		3*β*,5*α*,9*α*-Trihydroxy-(22*E*,24*R*)-ergosta-7,22-dien-6-one	*Phomopsis* sp. YM 355364	*A*. *carmichaeli* (T)	Antifungal	[71]
217		Phomopsterone B	*Phomopsis* sp. TJ507A	*P*. *glaucus* (T)	Anti-inflammatory	[93]
218		Cyathisterol	*Phomopsis* sp. YM 355364	*A*. *carmichaelii* (T)	Antifungal	[26]
219	Macrolides	Sch-642305	*Phomopsis* sp. CMU-LMA	*Alpinia malaccensis* (T)	Cytotoxic, Antimicrobial	[94]
220		LMA-P1	*Phomopsis* sp. CMU-LMA	*A*. *malaccensis* (T)	Cytotoxic	[94]
221		Benquoine	*Phomopsis* sp. CMU-LMA	*A*. *malaccensis* (T)	Cytotoxic, Antimicrobial	[94]
222		Aspergillide C	*Phomopsis* sp. IFB-ZS1-S4	*S*. *hainanensis* (M)	Inhibiting neuraminidase	[37]
223		Lithocarpin A	*P*. *lithocarpus* FS508	Sediment (M)	Cytotoxic	[95]
224		Lithocarpin B	*P*. *lithocarpus* FS508	Sediment (M)	Cytotoxic	[95]
225		Lithocarpin C	*P*. *lithocarpus* FS508	Sediment (M)	Cytotoxic	[95]
226		Lithocarpin D	*P*. *lithocarpus* FS508	Sediment (M)	Cytotoxic	[95]
227	Alkaloids	Phomopchalasin B	*Phomopsis* sp. shj2	*I. eriocalyx* var*. laxiflora* (T)	Antimigratory	[96]
228		Phomopsichalasin G	*P*. spp. xy21 and xy22	*X*. *granatum* (M)	Cytotoxic	[97]
229		18-Metoxycytochalasin J	*Phomopsis* sp.	*Garcinia kola* (T)	Cytotoxic, Antibacterial	[98]
230		Cytochalasin H	*Phomopsis* sp. *Phomopsis* sp. By254 *Phomopsis* sp.	*G*. *kola* (T) *Gossypium hirsutum* (T) *S*. *spectabilis* (T)	Cytotoxic, Antibacterial Antifungal Inhibiting acetylcholinesterase, Anti-inflammatory	[98] [99] [49]
231		Cytochalasin J	*Phomopsis* sp. *Phomopsis* sp. *P*. *asparagi*	*G*. *kola* (T) *S*. *spectabilis* (T) *Peperomia sui* (T)	Cytotoxic, Antibacterial Anti-inflammatory Antiandrogen	[98] [49] [100]
232		Phomopchalasin C	*Phomopsis* sp. shj2	*I. eriocalyx* var*. laxiflora* (T)	Cytotoxic, Anti-inflammatory, Antimigratory	[96]
233		Cytochalasin N	*Phomopsis* sp. By254	*G*. *hirsutum* (T)	Antifungal	[99]
234		Epoxycytochalasin H	*Phomopsis* sp. By254	*G*. *hirsutum* (T)	Antifungal	[99]
235		Diaporthalasin	*Phomopsis* sp. PSU-H188	*H*. *brasiliensis* (T)	Anti-MRSA	[73]
236		(+)-Tersone E	*P*. *tersa* FS441	Sediment (M)	Antibacterial, Cytotoxic	[101]
237		*ent*-Citridone A	*P*. *tersa* FS441	Sediment (M)	Antibacterial	[101]
238		Phochrodine C	*Phomopsis* sp. 33#	*R*. *stylosa* (M)	Anti-inflammatory	[102]
239		Phochrodine D	*Phomopsis* sp. 33#	*R*. *stylosa* (M)	Anti-inflammatory, Antioxidant	[102]
240		PM181110	*P*. *glabrae*	*Pongamia pinnata* (T)	Anticancer	[103]
241		Fusaristatin A	*P*. *longicolla* S1B4	- ^b^	Antibacterial	[34]
242		Exumolide A	*Phomopsis* sp. (No. ZH-111)	Sediment (M)	Accelerating the growth of SIV branch, Cytotoxic	[44]
243	Flavonoids	Quercetin	*P. castaneae-mollissimae* GQH87	*A*. *annua* (T)	Cytotoxic	[87]
244		Luteolin	*P. castaneae-mollissimae* GQH87	*A*. *annua* (T)	Cytotoxic	[87]
245		Naringenin	*P. castaneae-mollissimae* GQH87	*A*. *annua* (T)	Cytotoxic	[87]
246		Luteolin-7-*O*-glucoside	*P. castaneae-mollissimae* GQH87	*A*. *annua* (T)	Cytotoxic	[87]

^a^ T: terrestrial environment; M: marine environment; ^b^ The habitat was not mentioned.

**Table 2 microorganisms-09-00217-t002:** The bioactive secondary metabolites of the genus *Diaporthe* during 2010–2019.

Number	Structural Types	Compounds	Strains	Habitats (T/M ^a^)	Activities	Refs.
247	Xanthones	3,8-Dihydroxy-6-methyl-9-oxo-9*H*-xanthene-1-carboxylate	*Diaporthe* sp. SCSIO 41011	*Rhizophora stylosa* (M)	Anti-IAV	[125]
28		Phomoxanthone A	*Diaporthe* sp. GZU-1021 *D*. *phaseolorum* FS431	*Chiromanteshae-* *matochir* (M)Sediment (M)	Anti-inflammatory Cytotoxic	[126] [127]
248	Chromones	Penialidin A	*Diaporthe* sp. GZU-1021	*Chiromanteshae* *matochir* (M)	Anti-inflammatory	[126]
35		(+)-Phomopsichin A	*D*. *phaseolorum* SKS019	*Acanthus ilicifolius* (M)	Inhibitory effects on osteoclastogenesis	[128]
249		(−)-Phomopsichin A	*D*. *phaseolorum* SKS019	*A*. *ilicifolius* (M)	Inhibitory effects on osteoclastogenesis	[128]
250		(+)-Phomopsichin B	*D*. *phaseolorum* SKS019	*A*. *ilicifolius* (M)	Inhibitory effects on osteoclastogenesis	[128]
36		(−)-Phomopsichin B	*D*. *phaseolorum* SKS019*Diaporthe* sp. GZU-1021	*A*. *ilicifolius* (M) *Chiromateshaem* *atochir* (M)	Inhibitory effects on osteoclastogenesis Anti-inflammatory	[128] [126]
251		Diaporchromanone C	*D*. *phaseolorum* SKS019	*A*. *ilicifolius* (M)	Inhibitory effects on osteoclastogenesis	[128]
252		Diaporchromanone D	*D*. *phaseolorum* SKS019	*A*. *ilicifolius* (M)	Inhibitory effects on osteoclastogenesis	[128]
40		Pestalotiopsone F	*Diaporthe* sp. SCSIO 41011	*R*. *stylosa* (M)	Anti-IAV	[125]
253		Pestalotiopsone B	*Diaporthe* sp. SCSIO 41011 *D. pseudomangiferaea*	*R*. *stylosa* (M) *Tylophora ouata* (T)	Anti-IAV Antifibrotic	[125] [129]
254		Diaportheone A	*Diaporthe* sp. P133	*Pandanus amaryllifolius* (T)	Antitubercular	[130]
255		Diaportheone B	*Diaporthe* sp. P133	*P*. *amaryllifolius* (T)	Antitubercular	[130]
53	Chromanones	(10*S*)-Diaporthin	*D*. *terebinthifolii* LGMF907	*Schinus terebinthifolius* (T)	Antibacterial	[131]
256		Orthosporin	*D*. *terebinthifolii* LGMF907	*S*. *terebinthifolius* (T)	Antibacterial	[131]
54		Cytosporone D	*D. pseudomangiferaea*	*T*. *ouata* (T)	Cytotoxic, Antioxidant Antidiabetic	[129]
257		Mucorisocoumarin A	*D. pseudomangiferaea*	*T*. *ouata* (T)	Antifibrotic	[129]
258		3,4-Dihydro-8-hydroxy-3,5-dimethyl-isocoumarin	*D*. *eres*	*Hedera helix* (T)	Phytotoxic	[132]
259		Diportharine A	*Diaporthe* sp.	*Datura inoxia* (T)	Antioxidant	[133]
260	Furanones	(1*R*,2*E*,4*S*,5*R*)-1-[(2*R*)-5-Oxotetrahydrofuran-2-yl]-4,5-dihydroxy-hex-2-en-1-yl(2*E*)-2-methylbut-2-enoate	*Diaporthe* sp. SXZ-19	*Camptotheca acuminate* (T)	Cytotoxic	[134]
261		Butyl 5-[(1*R*)-1-hydroxyethyl]-γ-oxofuran-2-butanoate	*Diaporthe* sp. SXZ-19	*C*. *acuminate* (T)	Cytotoxic	[134]
262		3,4-Dihydro-5ʹ-[(1*R*)-1-hydroxyethyl] [2,2ʹ-bifuran]-5(2*H*)-one	*Diaporthe* sp. SXZ-19	*C*. *acuminate* (T)	Cytotoxic	[134]
263		3,4-Dihydro-5ʹ-[(1*R*)-1-hydroxymethylethyl][2,2ʹ-bifuran]-5(2*H*)-one	*Diaporthe* sp. SXZ-19	*C*. *acuminate* (T)	Cytotoxic	[134]
264		Kongiidiazadione	*D*. *Kongii*	*Carthamus lanatus* (T)	Phytotoxic, Antibacterial	[135]
265	Pyrones	Phomopsolide A	*D. maritima*	*Picea mariana*(T) *Picea rubens* (T)	Antifungal, Antibiotic	[136]
169		Phomopsolide B	*D. maritima*	*P*. *mariana* (T) *P*. *rubens* (T)	Antifungal, Antibiotic	[136]
266		Phomopsolide C	*D. maritima*	*P*. *mariana* (T)*P*. *rubens* (T)	Antifungal, Antibiotic	[136]
267		(*S*,*E*)-6-(4-Hydroxy-3-oxopent-1-en-1-yl)-2*H*-pyran-2-one	*D. maritima*	*P*. *mariana* (T) *P*. *rubens* (T)	Antifungal, Antibiotic	[136]
268		7-Hydroxy-6-metoxycoumarin	*D. lithocarpus*	*Artocarpus heterophyllus* (T)	Antifungal	[137]
269		Coumarin	*D. lithocarpus*	*A*. *heterophyllus* (T)	Antibacterial	[137]
270	Quinones	Phyllostine acetate	*D*. *miriciae*	*Cyperus iria* (T)	Antifeedant, Contact toxicity, Oviposition deterrent activities	[138]
107		(−)-Phyllostine	*D*. *miriciae*	*C*. *iria* (T)	Antifeedant, Contact toxicity, Oviposition deterrent activities	[138]
271		Biatriosporin N	*Diaporthe* sp. GZU-1021	*Chiromanteshae-* *matochir* (M)	Anti-inflammatory	[126]
272		Emodin	*D*. *lithocarpus*	*A*. *heterophyllus* (T)	Cytotoxic, Antibacterial	[137]
273		1,2,8-Trihydroxyanthraquinone	*D*. *lithocarpus*	*A*. *heterophyllus* (T)	Antibacterial	[137]
274		(+)-2,2′-Epicytoskyrin A	*Diaporthe* sp. GNBP-10	*Uncaria gambir* Roxb (T)	Antifungal	[139]
275		Cytoskyrin C	*Diaporthe* sp.	*Anoectochilus roxburghii* (T)	Cytotoxic, Activating the NF-κB pathway	[140]
276		(+)-Epicytoskyrin	*Diaporthe* sp.	*A*. *roxburghii* (T)	Cytotoxic, Activating the NF-κB pathway	[140]
277	Phenols	Tyrosol	*D*. *helianthin* *D*. *eres*	*Luehea divaricate* (T) *Vitis vinifera* (T)	Antagonistic Phytotoxic	[141] [142]
278		2,5-Dihydroxybenzyl alcohol	*D*. *vochysiae* LGMF1583	*Vochysia divergens* (T)	Cytotoxic	[143]
140		4-Hydroxybenzaldehyde	*D. eres*	*V*. *vinifera* (T)	Phytotoxic	[142]
279		*p*-Cresol	*D. eres*	*V*. *vinifera* (T)	Phytotoxic	[142]
280		4-Hydroxybenzoic acid	*D. eres*	*V*. *vinifera* (T)	Phytotoxic	[142]
281		Arbutin	*D. lithocarpus*	*A*. *heterophyllus* (T)	Cytotoxic	[137]
113		Phomosine A	*Diaporthe* sp. F2934	*Siparuna gesnerioides* (T)	Antibacterial	[144]
115		Phomosine C	*Diaporthe* sp. F2934	*S*. *gesnerioides* (T)	Antibacterial	[144]
282		Flavomannin-6,6′-di-*O*-methyl ether	*D*. *melonis*	*Annona squamosal* (T)	Antimicrobial	[145]
283		Acetoxydothiorelone B	*D. pseudomangiferaea*	*T*. *ouata* (T)	Antifibrotic	[129]
284		Dothiorelone B	*D. pseudomangiferaea*	*T*. *ouata* (T)	Antifibrotic	[129]
285		Dothiorelone L	*D. pseudomangiferaea*	*T*. *ouata* (T)	Antifibrotic	[129]
286		Dothiorelone G	*D. pseudomangiferaea*	*T*. *ouata* (T)	Antifibrotic	[129]
287		Diaporthol A	*Diaporthe* sp. ECN-137	*Phellodendron amurense* (T)	Anti-migration	[146]
288		Diaporthol B	*Diaporthe* sp. ECN-137	*P*. *amurense* (T)	Anti-migration	[146]
289		Tenellone C	*Diaporthe* sp. SYSU-HQ3	*Excoecaria agallocha* (M)	MptpB inhibitory	[147]
290		Tenellone D	*Diaporthe* sp. SYSU-HQ3	*E*. *agallocha* (M)	Anti-inflammatory	[148]
291		Diaporindene A	*Diaporthe* sp. SYSU-HQ3	*E*. *agallocha* (M)	Anti-inflammatory	[148]
292		Diaporindene B	*Diaporthe* sp. SYSU-HQ3	*E*. *agallocha* (M)	Anti-inflammatory	[148]
293		Diaporindene C	*Diaporthe* sp. SYSU-HQ3	*E*. *agallocha* (M)	Anti-inflammatory	[148]
294		Diaporindene D	*Diaporthe* sp. SYSU-HQ3	*E*. *agallocha* (M)	Anti-inflammatory	[148]
75		Isoprenylisobenzofuran A	*Diaporthe* sp. SYSU-HQ3	*E*. *agallocha* (M)	Anti-inflammatory	[148]
295	Oblongolides	Oblongolide D	*Diaporthe* sp. SXZ-19	*C*. *acuminate* (T)	Cytotoxic	[134]
296		Oblongolide H	*Diaporthe* sp. SXZ-19	*C*. *acuminate* (T)	Cytotoxic	[134]
297		Oblongolide P	*Diaporthe* sp. SXZ-19	*C*. *acuminate* (T)	Cytotoxic	[134]
298		Oblongolide V	*Diaporthe* sp. SXZ-19	*C*. *acuminate* (T)	Cytotoxic	[134]
299	Unclassified polyketides	Phomentrioloxin B	*D. gulyae*	*C*. *lanatus* (T)	Phytotoxic	[149]
300		*epi*-Isochromophilone II	*Diaporthe* sp. SCSIO 41011	*R*. *stylosa* (M)	Cytotoxic	[150]
301		Isochromophilone D	*Diaporthe* sp. SCSIO 41011	*R*. *stylosa* (M)	Cytotoxic	[150]
302	Monoterpenoids	(1*R*,2*R*,4*R*)-Trihydroxy-*p*-menthane	*Diaporthe* sp. SXZ-19	*C*. *acuminate* (T)	Cytotoxic	[134]
303		Gulypyrone A	*D*. *gulyae*	*C*. *lanatus* (T)	Phytotoxic	[149]
304		Gulypyrone B	*D*. *gulyae*	*C*. *lanatus* (T)	Phytotoxic	[149]
173		Nectriapyrone	*D. Kongii*	*C*. *lanatus* (T)	Phytotoxic	[135]
305	Sesquiterpenoids	Diaporol R	*Diaporthe* sp.	*R*. *stylosa* (M)	Cytotoxic	[151]
306		Eremofortin F	*Diaporthe* sp. SNB-GSS10	*Sabicea cinerea* (T)	Cytotoxic	[152]
307		Lithocarin B	*D*. *lithocarpus* A740	*Morinda officinalis* (T)	Cytotoxic	[153]
308		Lithocarin C	*D*. *lithocarpus* A740	*M*. *officinalis* (T)	Cytotoxic	[153]
309	Triterpenoids	19-Nor-lanosta-5(10),6,8,24-tetraene-1*α*,3*β*,12*β*,22*S*-tetraol	*Diaporthe* sp. LG23	*Mahonia fortunei* (T)	Antibacterial	[154]
216	Steriods	3*β*,5*α*,9*α*-Trihydroxy-(22*E*,24*R*)-ergosta-7,22-dien-6-one	*Diaporthe* sp. LG23	*M*. *fortunei* (T)	Antibacterial	[154]
310		Chaxine C	*Diaporthe* sp. LG23	*M*. *fortunei* (T)	Antibacterial	[154]
311	Ten-membered lactones	Phomolide C	*Diaporthe* sp.	*Aucuba japonica* var*. borealis* (T)	Inhibitory of proliferation of human colon adenocarcinoma cells	[155]
312		Xylarolide	*D. terebinthifolii*	*Glycyrrhiza glabra* (T)	Antimicrobial, Cytotoxic	[156]
313		Phomolide G	*D. terebinthifolii*	*G*. *glabra* (T)	Antibacterial	[156]
314		Xylarolide A	*Diaporthe* sp.	*D*. *inoxia* (T)	Cytotoxic, Antioxidant	[133]
315	Alkaloids	18-Des-hydroxy cytochalasin H	*D. phaseolorum*-92C	*Combretum lanceolatum* (T)	Inhibiting leishmanicidal, Antioxidant, Cytotoxic	[157]
316		21-Acetoxycytochalasin J_2_	*Diaporthe* sp. GDG-118	*Sophora tonkinensis* (T)	Antifungal, Antibacterial	[158]
317		21-Acetoxycytochalasin J_3_	*Diaporthe* sp. GDG-118	*S*. *tonkinensis* (T)	Antifungal, Antibacterial	[158]
318		Cytochalasin J_3_	*Diaporthe* sp. GDG-118	*S*. *tonkinensis* (T)	Antifungal, Antibacterial	[158]
230		Cytochalasin H	*Diaporthe* sp. GDG-118 *Diaporthe* sp. GZU-1021	*S*. *tonkinensis* (T) *Chiromanteshae* *matochir* (M)	Antifungal, AntibacterialAnti-inflammatory	[158] [126]
319		7-Acetoxycytochalasin H	*Diaporthe* sp. GDG-118	*S*. *tonkinensis* (T)	Antifungal, Antibacterial	[158]
231		Cytochalasin J	*Diaporthe* sp. GDG-118	*S*. *tonkinensis* (T)	Antifungal, Antibacterial	[158]
320		Cytochalasin E	*Diaporthe* sp. GDG-118	*S*. *tonkinensis* (T)	Antifungal, Antibacterial	[158]
321		21-*O*-Deacetyl-L-696,474	*Diaporthe* sp. GZU-1021	*Chiromanteshae* *matochir* (M)	Anti-inflammatory	[126]
322		Cordysinin A	*D. arecae*	*Kandelia obovate* (M)	Anti-angiogenic	[159]
323		5-Deoxybostrycoidin	*D. phaseolorum* SKS019	*A*. *ilicifolius* (M)	Cytotoxic	[160]
241		Fusaristatin A	*D. phaseolorum* SKS019	*A*. *ilicifolius* (M)	Cytotoxic	[160]
324		Vochysiamide B	*D. vochysiae* LGMF1583	*V*. *divergens* (T)	Antibacterial, Cytotoxic	[143]
325		Diaporisoindole A	*Diaporthe* sp. SYSU-HQ3	*E*. *agallocha* (M)	Anti-inflammatory	[148]
326		Diaporisoindole B	*Diaporthe* sp. SYSU-HQ3	*E*. *agallocha* (M)	Anti-inflammatory	[148]
327		Diaporisoindole D	*Diaporthe* sp. SYSU-HQ3 *Diaporthe* sp. SYSU-HQ3	*E*. *agallocha* (M) *E*. *agallocha* (M)	Anti-inflammatory MptpB inhibitory	[148] [147]
328		Diaporisoindole E	*Diaporthe* sp. SYSU-HQ3	*E*. *agallocha* (M)	Anti-inflammatory	[148]
329		Phomopsin F	*D. toxica*	*-* ^b^	Cytotoxic	[161]
330	Fatty acids	3-Hydroxypropionic acid	*D. phaseolorum*	*Laguncularia racemose* (M)	Antimicrobial	[162]
331		3-Nitropropionic acid	*D*. *gulyae*	*C*. *lanatus* (T)	Phytotoxic	[149]
332		Diapolic acid A	*D. terebinthifolii*	*G*. *glabra* (T)	Antibacterial	[156]
333		Diapolic acid B	*D. terebinthifolii*	*G*. *glabra* (T)	Antibacterial	[156]
334		Diaporthsin E	*Diaporthe* sp. JC-J7	*Dendrobium nobile* (T)	Antihyperlipidemic	[163]
335		3-Hydroxy-5-methoxyhex-5-ene-2,4-dione	*Diaporthe* sp. ED2	*Orthosiphon stamieus* (T)	Antifungal	[164]

^a^ T: terrestrial environment; M: marine environment; ^b^ The habitat was not mentioned.

## Data Availability

All data in this article is openly available without any restrictions.

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
