# Peer review of "Bioactive Secondary Metabolites of the Genus Diaporthe and Anamorph Phomopsis from Terrestrial and Marine Habitats and Endophytes: 2010–2019"

_microorganisms, 2021, doi:10.3390/microorganisms9020217_

Round 1
Reviewer 1 Report
This is a good writen review concerning extensive list of bioactive secondary metabolites discovered from Phomopsis and Diaporthe during 2010-2019, and their bioactivities. However, I think it's required some adjustments and correction.
- The title should be more precise , is this review concerns the terrestial and marine habitats and endophytes recovered from that ecosystems, becouse it is confusing. Reading the Introduction, the authors claim that they are describing both, but it is not clear. Moreover in the conclusions part, lines 1035-1037, there is information about percentage distribution compounds obtained from the terrestrial and marine environment, but I was not able to find such data in table 1. I think it should be highlited by histogram, to make this data more clear and visable.
- The citation of latin names of hosts in Table 1 and 2 is confusing. Generally the name first time citated is given as well the genus name as well as the species name, it the following should be used abbreviation of genus name. So in line 25 should be the whole name , while in 26 the abbreviation of genus host plant. In line 28 should be the compleated, whole name of habitat, please correct it as following in both tables.
- There are ovelaps, I think it should be skip the informations concerning metabolites and their bioactivities in the text, while they are given in tabs.,for exemple data in lines 356-358. Please remove such repetitions, it makes the manuscript shorter and more friendly to read.
- There are aslo overlaps in Conclusions part and Introduction: data in lines 1062-1063 demonstrate the same data like in lines 63-64.
Reviewer 2 Report
It is a very interesting review where the researchers summarized the bioactive secondary metabolites identified in Diaporthe and Phomopsis during 2010-2019. It is very well organized and explained.
There are only some mistakes:
Keyword: Change them (yellow color) because the keyworks cannot be in the title. Diaporthe; Phomopsis; endophytic fungi; plant pathogens; biological activities; secondary metabolites
The authority for a Latin binomial name should be provided after each common name the first time it is referred to in the title, abstract, main body, and a figure or table description.
When a species is cited for the first time in the text, the full text should be written for example Paris polyphylla var. yunnanensis. But afterwards it is abbreviated, as for example P polyphylla var. yunnanensis. Correct “var.” is without italic.
In vitro and in vivo must write in italics in all text. Correct in all text.
Tables: Add a column with the number of each compound. Put a mark on the table when it changes structural types for easy compression and viewing. If this would be possible, add a new column with the molecular weight in each compound.
Correct: habitat of the compound 102.
When it is written, for example lines 606, (this is repeated in all text) “A549, MDA-MB-231, and PANC-1”, what are they? Bacteria, fungus, isolates, cancer cells, ??? Explain it in the text. (all text).
Figure 20: Check it. I think the percentages are incorrect.
Lines 1048-1075 and 1067-1069: These lines are not conclusions.
Line 1127: “Leguminosae” is without italic.
Line 1179: “Isolated” is in small letter.
